# Endothelial SIRPα signaling controls VE-cadherin endocytosis for thymic homing of progenitor cells

**Boyang Ren[1,2], Huan Xia[1,2], Yijun Liao[1,2], Hang Zhou[1,2], Zhongnan Wang[1], Yaoyao Shi[1], Mingzhao Zhu[1,2]***

[1]The Key Laboratory of Infection and Immunity, Institute of Biophysics, Chinese Academy of Sciences, Beijing, China; [2]College of Life Sciences, University of the Chinese Academy of Sciences, Beijing, China

**Abstract** Thymic homing of hematopoietic progenitor cells (HPCs) is tightly regulated for proper T cell development. Previously we have identified a subset of specialized thymic portal endothelial cells (TPECs), which is important for thymic HPC homing. However, the underlying molecular mechanism still remains unknown. Here, we found that signal regulatory protein alpha (SIRPα) is preferentially expressed on TPECs. Disruption of CD47-SIRPα signaling in mice resulted in reduced number of thymic early T cell progenitors (ETPs), impaired thymic HPC homing, and altered early development of thymocytes. Mechanistically, *Sirpa*-deficient ECs and *Cd47*-deficient bone marrow progenitor cells or T lymphocytes demonstrated impaired transendothelial migration (TEM). Specifically, SIRPα intracellular ITIM motif-initiated downstream signaling in ECs was found to be required for TEM in an SHP2- and Src-dependent manner. Furthermore, CD47 signaling from migrating cells and SIRPα intracellular signaling were found to be required for VE-cadherin endocytosis in ECs. Thus, our study reveals a novel role of endothelial SIRPα signaling for thymic HPC homing for T cell development.

*For correspondence:
zhumz@ibp.ac.cn

Competing interest: The authors declare that no competing interests exist.

## Editor's evaluation

The primary audience that will be keenly interested in these findings will be those with an interest in T cell development and thymic function, however given the broad applicability of transendothelial migration, there will also likely be broader interest in these findings. The work provides key new insights as to the role of the CD47-SIRPa signaling axis in the regulation of how hematopoietic progenitors enter into the thymus to initiate T cell development.

## Introduction

Different from most other hematopoietic cells, T cells develop in the thymus. Thymic homing of bone marrow-derived hematopoietic progenitor cells (HPCs) is therefore a critical step. It was reported that HPCs enter the thymus via unique blood vessels that are surrounded by perivascular spaces (PVS) (*Lind et al., 2001*; *Mori et al., 2007*) primarily in the corticomedullary junction (CMJ) of the thymus. To understand how the unique blood vascular endothelial cells (ECs) are involved in thymic homing of HPCs is an important question in the field. Previously, we identified a specialized subset of P-selectin+Ly6C- thymic portal endothelial cells (TPECs) located in the CMJ region and associated with PVS structure, providing the cellular basis for thymic homing of HPCs (*Shi et al., 2016*). Short-term HPC thymic homing assay confirmed that TPECs is highly associated with settling HPCs and lack of TPECs in lymphotoxin beta receptor-deficient mice resulted in dramatically impaired HPC thymic homing

and reduced number of thymic early T cell progenitors (ETPs). Transcriptome analysis revealed that TPECs are enriched with transcripts related to cell adhesion and trafficking, supporting its critical role for thymic homing. However, the molecular basis of TPEC function and its underlying mechanism have not been experimentally determined.

Several molecules have been found to play important roles for thymic homing of HPCs. P-selectin and adhesion molecule VCAM-1 and ICAM-1, highly expressed on ECs, as also confirmed in our previous study, have been suggested to mediate adhesion of HPCs on ECs (*Lind et al., 2001*; *Mori et al., 2007*; *Rossi et al., 2005*; *Scimone et al., 2006*). In addition, chemokines such as CCL25 and CCL19/CCL21 expressed by thymic ECs as well as thymic epithelial cells (TECs) are also involved in thymic homing of HPCs, probably via integrin activation (*Krueger et al., 2010*; *Misslitz et al., 2004*; *Parmo-Cabañas et al., 2007*; *Zhang et al., 2014*; *Zlotoff et al., 2010*). All the above-mentioned mechanisms regulate the multistep HPC homing at early invertible process (*Zlotoff and Bhandoola, 2011*). Transendothelial migration (TEM) is the decisive step for migration of progenitors from blood into the thymus (*Zlotoff and Bhandoola, 2011*). To accomplish the homing process, the barrier of endothelial junction must be opened. How this step is regulated in TPECs still remains unknown.

Signal regulatory protein alpha (SIRPα) is a transmembrane protein that contains three Ig-like domains, a single transmembrane region, and a cytoplasmic region. Ligation of SIRPα by its ligand CD47 transmits intracellular signal through its ITIM motifs. SIRPα is mainly expressed by myeloid cells such as monocytes, granulocytes, most tissue macrophages, and subsets of dendritic cells (*Barclay and Van den Berg, 2014*). On the other hand, CD47 is ubiquitously expressed but show fluctuating expression levels on different cell states or cell types. Elevated CD47 expression is detected on bone marrow cells and some of lymphoid subsets (*Jaiswal et al., 2009*; *Van et al., 2012*).

SIRPα plays various roles in immune system. SIRPα on conventional dendritic cells (cDCs) maintains their survival and proper function via intracellular SHP2 signaling (*Iwamura et al., 2011*; *Saito et al., 2010*). On macrophages, upon CD47 ligation, SIRPα signaling activates intracellular SHP1 to inhibit Fcγ receptor-mediated phagocytosis toward target cells (*Blazar et al., 2001*; *Ishikawa-Sekigami et al., 2006*; *Tsai and Discher, 2008*). Elevated expression of CD47 on platelets, lymphocyte subsets, and hematopoietic cell subsets shows 'self' identity and protects them from being cleared by macrophages (*Blazar et al., 2001*; *Jaiswal et al., 2009*; *Olsson et al., 2005*; *Yamao et al., 2002*). Tumor cells express high level of CD47 to escape immune surveillance initiated by macrophages (*Chao et al., 2010*; *Majeti et al., 2009*; *Willingham et al., 2012*), which leads to the development of CD47-SIRPα blockade as a promising approach for cancer therapy (*Feng et al., 2019*).

CD47-SIRPα has also been reported to play important roles in cell adhesion and migration. SIRPα expressed on cDCs, neutrophils, melanoma cells, and CHO cells has been reported to promote cell motility (*Fukunaga et al., 2004*; *Liu et al., 2002*; *Motegi et al., 2003*) through SHP2-dependent activation of Rho GTPase (*Wollenberg et al., 1996*) and cytoskeleton reorganization (*Inagaki et al., 2000*). On the other hand, SIRPα expressed on neutrophils and monocytes has been shown to be the ligand for endothelial or epithelial CD47 and promotes junction opening through activating Rho family GTPase therefore permitting transmigration of the SIRPα-expressing cells (*de Vries et al., 2002*; *Liu et al., 2002*; *Stefanidakis et al., 2008*). However, the expression and function of SIRPα on ECs remain unknown.

In the present study, we uncovered a novel role of TPEC signature molecule SIRPα in thymic homing of progenitor cells, revealed that migrating cell-derived CD47 and EC-SIRPα intracellular signal induce junctional VE-cadherin endocytosis and promote TEM.

## Results

### TPECs preferentially express SIRPα

To explore the molecular mechanisms for thymic homing of the progenitors through TPECs, we started with signature genes of TPECs based on previously published RNA-Seq data (GSE_83114) and focused on the genes related to cell adhesion and migration by intersecting with related gene sets (*Shi et al., 2016*; *Supplementary file 1*). Among those signature genes of TPECs, SIRPα (*Figure 1A and B*) is of particular interest given its involvement in cell migration on leukocytes but undiscovered role on non-immune cells.

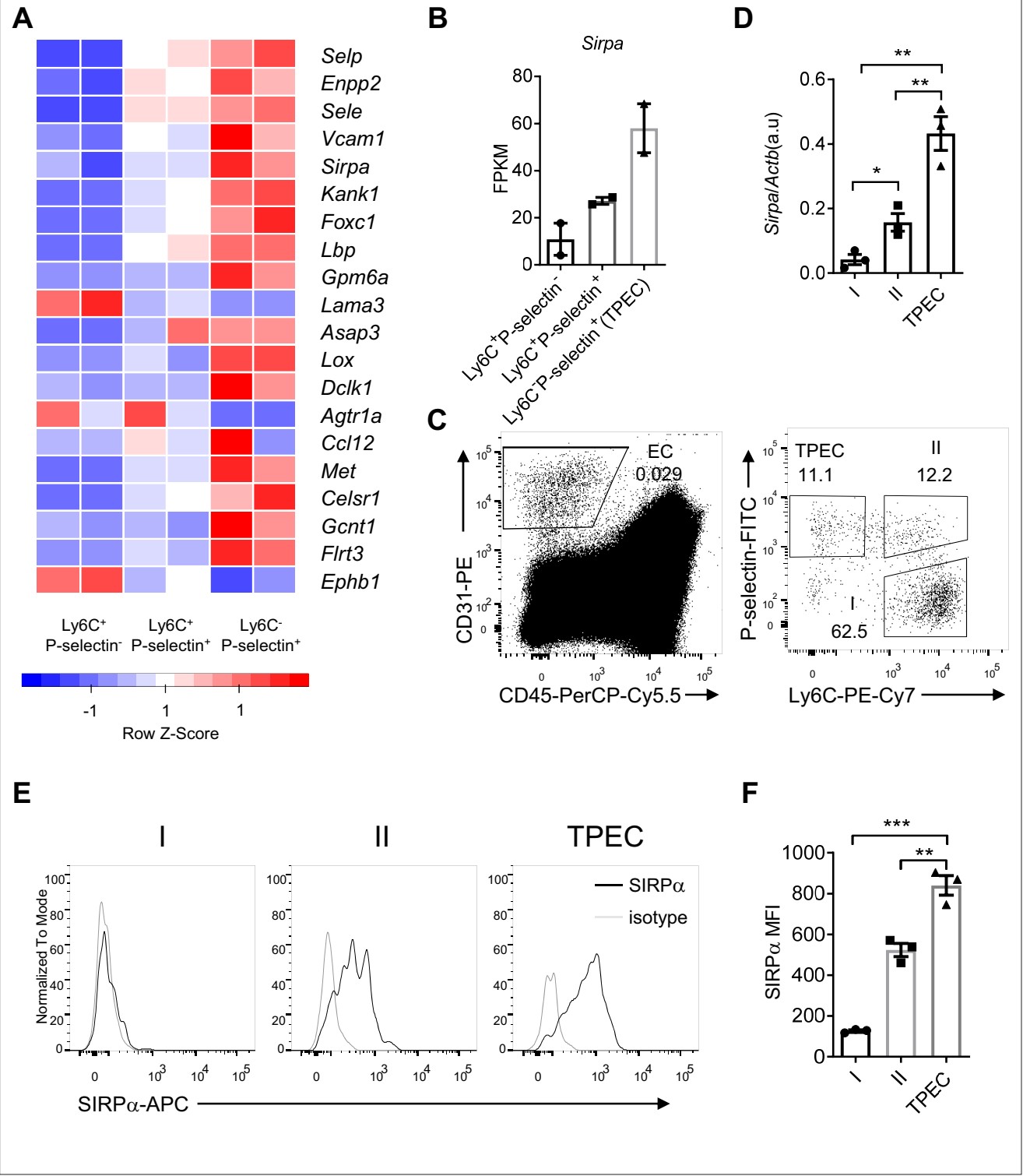

**Figure 1.** Signal regulatory protein alpha (SIRPα) is preferentially expressed on thymic portal endothelial cells (TPECs). (**A**) Expression profile of top 20 signature genes of Ly6C⁻Selp⁺ ECs (TPECs), which have absolute FC > 2 and p < 0.01 in TPECs versus either Ly6C⁺Selp⁻ or Ly6C⁺Selp⁺ thymic EC subsets and are in GO term GO_0016477 (cell migration). Relative expression of each gene among EC subsets are presented as mean-centered z-score distribution. (**B**) Expression level of *Sirpa* among the three thymic EC subsets. FPKM: fragments per kilobase per million mapped reads. (**C**) Flow cytometric analysis of thymic ECs (CD31⁺CD45⁻) and subset I (Ly6C⁺Selp⁻), subset II (Ly6C⁺Selp⁺), and subset III (TPEC, Ly6C⁻Selp⁺). (**D**) Real-time PCR analysis of *Sirpa* mRNA expression in thymic EC subsets, n = 3, each dot represents sample from individual mouse, data are representative of two independent experiments. (**E–F**) Flow cytometry analysis of SIRPα expression on the three thymic EC subsets (**E**) and quantification of measuring mean

*Figure 1 continued on next page*

*Figure 1 continued*

fluorescence intensity of SIRPα (**F**), data are representative of three independent experiments with three biological replicates (n=3) in each group. Error bars represent s.e.m. Asterisks mark statistically significant difference, *p < 0.05, **p < 0.01, and ***p < 0.001 determined by two-tailed unpaired Student's t-test. Source data and detailed method for generating heatmap in A are available in *Figure 1—source data 1*. FPKM table for GSE_83114 is available in *Supplementary file 1*.

The online version of this article includes the following source data and figure supplement(s) for figure 1:

**Source data 1.** Source data file for Figure 1.

**Figure supplement 1.** Signal regulatory protein alpha (SIRPα) expression on thymic portal endothelial cells (TPECs).

**Figure supplement 1—source data 1.** Source data file for Figure 1-figure supplement 1.

To further assess the expression of SIRPα on thymic endothelial subsets, CD31⁺ thymic ECs were separated by Ly6C and P-selectin (*Figure 1C*) as previously reported (*Shi et al., 2016*), and analyzed by quantitative RT-PCR and flow cytometry. SIRPα was barely detectable on Ly6C⁺P-selectin⁻ ECs, the dominant population of thymic ECs. On Ly6C⁺P-selectin⁺ ECs, the suggested precursor of TPECs, there was a substantial level of SIRPα expression. Among all three EC subsets, Ly6C⁻P-selectin⁺ TPECs express the highest level of SIRPα at both mRNA level (*Figure 1D*) and protein level (*Figure 1E and F*). Immunofluorescence staining also confirmed the location of SIRPα expression on TPECs, although irrespective of PVS (*Figure 1—figure supplement 1A, B*). Thus, SIRPα appears to be closely and positively related to thymic TPECs.

Given the preferential expression of SIRPα on TPECs, we asked whether signals affect TPEC development such as LT-LTβR would be required for SIRPα expression. However, the remaining TPECs in *Ltbr⁻ᐟ⁻* mice showed similar level of SIRPα expression compared to that in WT mice (*Figure 1—figure supplement 1C*). Thymic stromal niche is another factor regulating thymic settling of progenitor cells (*Krueger, 2018*; *Prockop and Petrie, 2004*). Increased thymic stromal niche was found during irradiation (*Zlotoff et al., 2011*). However, this does not seem to alter SIRPα expression, either (*Figure 1—figure supplement 1D*). Whether SIRPα expression on TPECs is a constitutive event or regulatable upon thymic microenvironmental change remains to be tested in future.

## SIRPα is essential for ETP population maintenance, thymic progenitor homing, and proper T cell development

ETPs are the very first subset of bone marrow-derived progenitor cells that settle the thymus and are committed to T cell lineage. Studies have suggested ETP population size a direct indicator of thymic HPC homing (*Krueger et al., 2010*; *Shi et al., 2016*; *Zlotoff et al., 2010*). Therefore, to test the requirement of SIRPα for thymic progenitor homing, we first analyzed ETP population in *Sirpa⁻ᐟ⁻* mice. *Sirpa⁻ᐟ⁻* mice appeared grossly normal in thymic development of T cells (*Figure 2—figure supplement 1A*). However, the frequencies and numbers of DN thymocytes were slightly reduced (*Figure 2—figure supplement 1B, C*). More specifically, the frequencies and numbers of DN1, DN2, and DN3 thymocytes all exhibited significant reduction (*Figure 2—figure supplement 1D, E*). The DN thymocyte deficiency started at DN1, which mainly consists of ETPs. A substantial reduction in ETP population was confirmed in *Sirpa⁻ᐟ⁻* mice (*Figure 2A and B*). Bone marrow multipotent progenitors including lineage⁻Sca-1⁺c-Kit^high^ (LSK) and common lymphoid progenitors (CLPs) remained unchanged in these mice (*Figure 2—figure supplement 1F, G*), suggesting the reduced population size of thymic ETP is unlikely due to impaired generation/homeostasis of bone marrow progenitors.

Previous studies suggested that thymic entry of the progenitor cells and ETP population maintenance is guided by multiple cues derived from thymic ECs as well as TECs. To specifically test the role of endothelial SIRPα, we generated *Sirpa*^flox/flox^;*Tek*^cre^ conditional knockout (*Sirpa* TEK-CKO) mice. Significant decrease of thymic ETP population was found in *Sirpa* TEK-CKO mice in a degree similar to that in *Sirpa⁻ᐟ⁻* mice (*Figure 2C and D*), suggesting the role of SIRPα is probably mainly derived from ECs. In supporting this, much lower expression level of SIRPα was found on either medullary or cortical TECs (*Figure 2—figure supplement 2A-C*).

*Sirpa* TEK-CKO also deleted *Sirpa* in hematopoietic lineage. Previous study has reported SIRPα as a phagocytic checkpoint on tissue macrophages and CD47-null cells are rapidly cleared in congenic wild type (WT) mice (*Bian et al., 2016*). Therefore, possibility exists that phagocytic activity may elevate toward circulating progenitor cells in *Sirpa⁻ᐟ⁻* or *Sirpa* TEK-CKO mice before they reach the

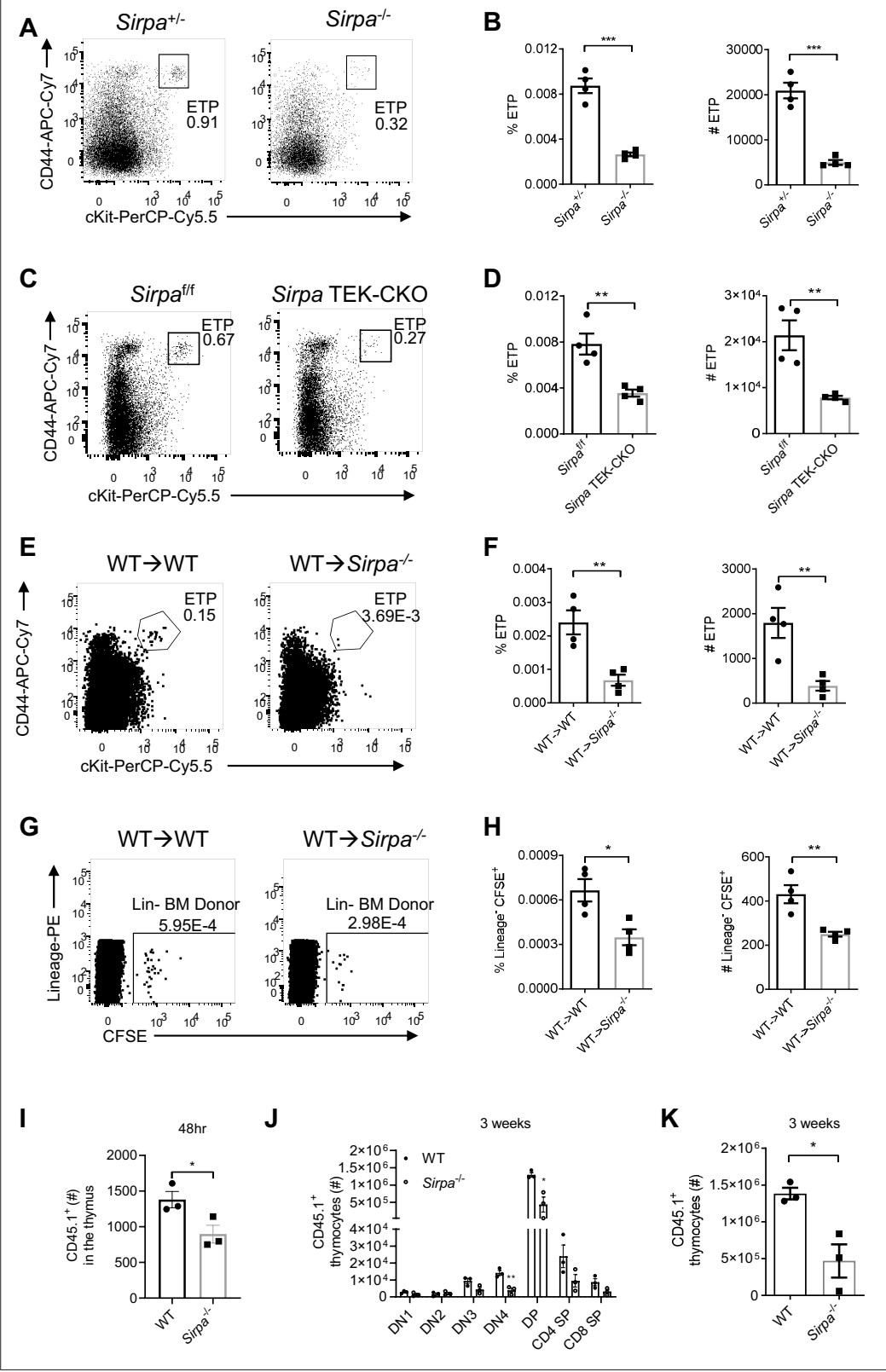

**Figure 2.** Signal regulatory protein alpha (SIRPα) is essential for early T cell progenitor (ETP) population maintenance, thymic progenitor homing, and proper T cell development. (**A**) Representative flow cytometric analysis of ETPs (lineage⁻CD25⁻CD44⁺cKit⁺) in the thymus of *Sirpa⁻/⁻* and control mice. (**B**) Proportion of ETP population of total thymocytes and corresponding cell number in a thymus. n = 4 in each group, data are

*Figure 2 continued on next page*

*Figure 2 continued*

representative of three independent experiments. (**C**) Flow cytometric analysis of ETPs in *Sirpa* TEK-CKO mice. (**D**) Statistics of ETPs in the thymus, n = 4 in each group, data are representative of three independent experiments. (**E**) Representative flow cytometric analysis of ETPs (lineage⁻CD25⁻CD44⁺cKit⁺) in wild type (WT) bone marrow reconstituted WT or *Sirpa*⁻/⁻ mice. (**F**) Statistics of ETPs in the thymus, n = 4 in each group, data are representative of two independent experiments. (**G,H**) Whole bone marrow short-term homing assay in WT bone marrow reconstituted (8 weeks after lethal irradiation and bone marrow transplant) WT or *Sirpa*⁻/⁻mice. (**G**) Analysis of lineage-negative donor cells (Lin⁻CFSE⁺) among total thymocytes. (**H**) Statistics of lineage-negative donor cells in the thymus. n = 4 in each group, data are representative of two independent experiments. (**I**) Statistics of Lin⁻ CD45.1⁺ donor cells in the thymus 48 hr after adoptive transfer. n = 3 in each group, data are representative of two independent experiments. (**J,K**) Statistics of donor-derived thymocyte subsets (**J**) and total CD45.1⁺ cell numbers (**K**) in the recipient thymus. Lin⁻ BMCs were transferred as in (**I**), 3 weeks later, donor-derived thymocytes were detected. n = 3 in each group, data are representative of two independent experiments. Error bars represent s.e.m. Asterisks mark statistically significant difference, \*$p < 0.05$, \*\*\*$p < 0.001$, \*\*\*\*$p < 0.0001$, n.s. not significant, determined by two-tailed unpaired Student's t-test. Source data are available in *Figure 2—source data 1*.

The online version of this article includes the following source data and figure supplement(s) for figure 2:

**Source data 1.** Source data file for Figure 2.

**Figure supplement 1.** Signal regulatory protein alpha (SIRPα) deficiency has minor effect on T cell development at steady state.

**Figure supplement 1—source data 1.** Source data file for Figure 2-figure supplement 1.

**Figure supplement 2.** Endothelial signal regulatory protein alpha (SIRPα) is essential for thymic progenitor homing.

**Figure supplement 2—source data 1.** Source data file for Figure 2-figure supplement 2.

---

thymus, since *Sirpa* is deficient in hematopoietic cells of both mouse lines. To distinguish the role of hematopoietic cell-derived versus non-hematopoietic radioresistant stromal cell-derived SIRPα in the regulation of thymic ETPs, bone marrow chimeric mice were generated (*Figure 2—figure supplement 2D*). Mice lacking SIRPα on radioresistant stromal cells demonstrated significantly reduced ETP population (*Figure 2E and F*), whereas in mice lacking SIRPα on hematopoietic cells, ETP population remained unchanged (*Figure 2—figure supplement 2E, F*). Thus, together with the previous data, reduced thymic ETP population in *Sirpa*⁻/⁻ or *Sirpa* TEK-CKO mice is unlikely due to increased clearance of HPCs in the absence of hematopoietic SIRPα. In future, an EC-specific deletion of SIRPα, such as with *Cdh5*-CreERT2 mice, may provide more conclusive information.

To directly test the role of SIRPα on HPC thymic homing, a short-term thymic homing assay was adopted (*Shi et al., 2016*), in which large amount of progenitor-containing bone marrow cells were intravenously transferred and thymic settling progenitor cells were determined 2 days later. To separate the hematopoietic versus non-hematopoietic role of SIRPα, bone marrow chimeric mice were used for short-term homing assay (*Figure 2—figure supplement 2D*). SIRPα deficiency on hematopoietic cells did not result in impaired thymic settling of bone marrow progenitors (*Figure 2—figure supplement 2G*), nor did the number of progenitors retained in the periphery as represented by the number of progenitor cells in the spleen (*Figure 2—figure supplement 2H*). On the contrary, loss of SIRPα on radioresistant cells significantly impeded thymic entry of progenitor cells (*Figure 2G and H*), while the number of donor bone marrow cells retained unaltered in the periphery (*Figure 2—figure supplement 2I*).

To avoid the potential influence of massive amount of bone marrow cells during short-term homing assay, or the potential effect of irradiation during the construction of bone marrow chimeric mice, we enriched lineage-negative bone marrow progenitor cells (Lin⁻ BMCs) and did short-term homing assay in naïve non-irradiated WT or *Sirpa*⁻/⁻ mice. 1 × 10⁶ Lin⁻ BMCs were used per mouse. In this more physiologically relevant setting, short-term homing of Lin⁻ BMCs still showed significant defect in *Sirpa*⁻/⁻ mice (*Figure 2I*). As a reference, donor progenitors remained the same in the spleen (*Figure 2—figure supplement 2J*), which further confirmed the role of endothelial SIRPα in controlling thymic entry of the progenitors.

To test whether the defective thymic homing of progenitor cells in *Sirpa*⁻/⁻ mice would influence T cell development, we examined donor-derived thymocytes 3 weeks after adoptive transfer. At this time point, donor-derived cells have developed through all major stages of thymocytes (*Figure 2J*),

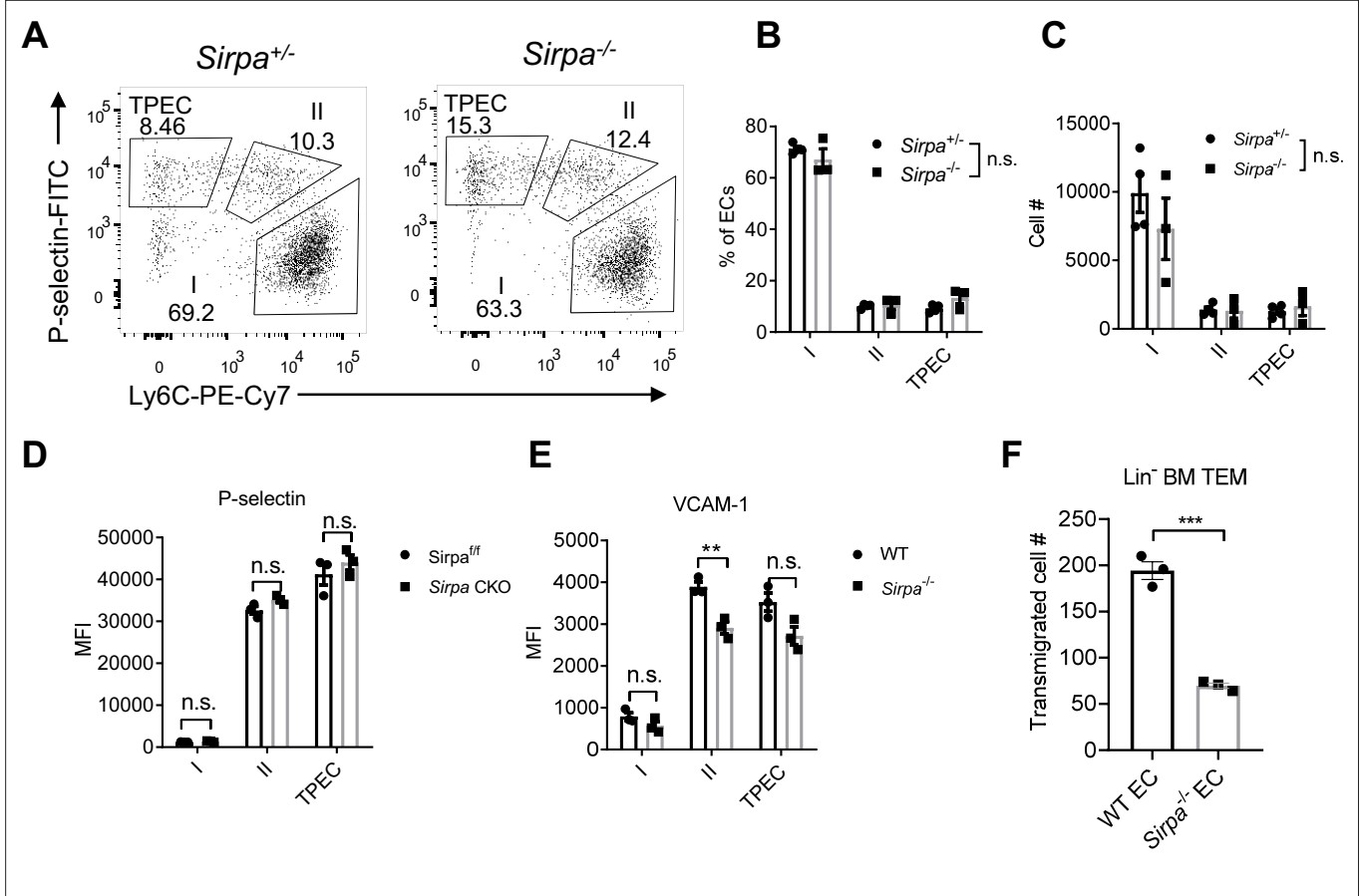

**Figure 3.** Endothelial cell (EC)-signal regulatory protein alpha (SIRPα) controls bone marrow progenitor cell transendothelial migration (TEM). (**A**) Representative flow cytometric analysis of thymic EC composition in *Sirpa⁻/⁻* and control mice. (**B,C**) Proportion of thymic EC subsets (**B**) and corresponding cell numbers (**C**) in the thymus, n = 4 in *Sirpa⁺/⁻* and n = 3 in *Sirpa⁻/⁻* group, data are representative of three independent experiments. (**D,E**) Expression level of adhesion molecules on thymic EC subsets. (**D**) P-selectin in *Sirpa* TEK-CKO or control mice, n = 3 for each group. (**E**) VCAM-1 in *Sirpa⁻/⁻* or control mice, n = 3 for each group, data are representative of three independent experiments. (**F**) FACS-sorted Lin⁻ BMC transmigration driven by CCL19, n = 3 in each group, data are representative of three independent experiments. Error bars represent s.e.m. Asterisks mark statistically significant difference, **p < 0.01, n.s. not significant, determined by two-tailed unpaired Student's t-test. Source data are available in *Figure 3—source data 1*.

The online version of this article includes the following source data and figure supplement(s) for figure 3:

**Source data 1.** Source data file for Figure 3.

**Figure supplement 1.** Signal regulatory protein alpha (SIRPα) does not control endothelial cell (EC) development and growth.

**Figure supplement 1—source data 1.** Source data file for Figure 3-figure supplement 1.

and significantly less repopulated in *Sirpa⁻/⁻* mice, especially for DP thymocytes (*Figure 2J*). The total thymocytes derived from donor progenitor cells is also reduced in *Sirpa⁻/⁻* mice (*Figure 2K*).

Together, these data suggest that myeloid-SIRPα-mediated phagocytic activity does not affect thymic progenitor homing in the current system; endothelial SIRPα plays an important role on thymic homing of progenitor cells, ETP population maintenance and proper T cell development.

## EC-SIRPα controls bone marrow progenitor cell TEM

Since TPECs is the portal of thymic progenitor entry, we asked whether SIRPα signaling may regulate TPEC development. *Sirpa⁻/⁻* mice did not show altered thymic endothelial development in regard to the percentage and number of total ECs (*Figure 3—figure supplement 1A-C*) and specifically TPECs (*Figure 3A–C*). Adhesion of progenitor cells to the wall of blood vessel is a multistep progress before transmigration toward thymic parenchyma, involving various selectins, adhesion molecules, and chemokines (*Scimone et al., 2006*; *Zlotoff and Bhandoola, 2011*). Since SIRPα has been reported for

its role in cellular adhesion (*Seiffert et al., 1999*), we next tested the key molecules involved in thymic progenitor adhesion. P-selectin, VCAM-1, *Ccl25,* and *Ccl19* showed no significant change on TPECs of *Sirpa*-/- mice (*Figure 3D and E*, *Figure 3—figure supplement 1D*). Only *Ccl21a* showed slight but significantly reduced expression in *Sirpa*-/- EC (*Figure 3—figure supplement 1D*). Considering unreduced thymic ETP population in CCR7-deficient mice, the reduced *Ccl21a* expression in *Sirpa*-/- EC unlikely explains the significant ETP reduction in *Sirpa*-/- mice. To test other potential unknown effect of endothelial SIRPα on leukocyte-EC adhesion, we directly tested this in vitro. Sirpa-/- MS1 EC line was constructed by CRISPR-Cas9 and the deletion of SIRPα was confirmed by flow cytometry (*Figure 3—figure supplement 1E*). Compared to WT ECs, *Sirpa*-/- EC monolayer mediates comparable adhesion of lymphocytes (*Figure 3—figure supplement 1F*).

Next, we examined the role of SIRPα in TEM by transwell assay. Monolayers of both WT and *Sirpa*-/- ECs were equally formed at the time of transmigration assay (*Figure 3—figure supplement 1G, H*). In the presence of CCL19 in the lower chambers, FACS-sorted Lin- BMCs were measured for their transmigration across endothelial barrier. Remarkably, SIRPα deficiency in ECs resulted in significant reduction of transmigrated cell numbers (*Figure 3F*), suggesting a direct role of endothelial SIRPα in controlling progenitor cell TEM. A similar effect was also found when naïve T lymphocytes were used for transwell assay (*Figure 3—figure supplement 1I*), suggesting a general mechanism of endothelial SIRPα in controlling TEM.

## Migrating cell-derived CD47 guides their TEM

CD47, the cellular ligand of SIRPα, is ubiquitously expressed on almost all types of cells. Interestingly, we found among developmental subsets of T cell lineage, ETPs exhibited the highest level of CD47 expression (*Figure 4A*, *Figure 4—figure supplement 1A*), suggesting preferential interaction may exist between ETPs and TPECs controlling thymic progenitor homing and ETP population. Moreover, FACS staining revealed higher level of CD47 expression on the more primitive Flt3hi ETP subset (*Figure 4B*, gating strategy: *Figure 4—figure supplement 1B*). Indeed, *Cd47*-/- mice showed significantly reduced ETP population (*Figure 4C–E*) and unaffected ancestral progenitor subsets (*Figure 4—figure supplement 1C-E*), similar to that of *Sirpa*-/- mice. To directly test the role of CD47 on migrating cells in TEM, the transwell assay was applied. It should be noted that significantly elevated CD47 expression was found on immortalized cells, herein MS1 (*Figure 4—figure supplement 1F*), which was not found in the physiological situation, wherein TPEC-CD47 expression level is much lower than that on migrating cells (ETP) (*Figure 4—figure supplement 1G, H*). Thus, although lymphocyte transmigration across *Cd47*-/- MS1 was significantly reduced compared to that on WT MS1, it probably does not reflect a physiological role (*Figure 4—figure supplement 1I, J*). To exclude the potential influence of this artificial CD47 expression on ECs and to determine the sole role of migrating cell-derived CD47, *Cd47*-/- MS1 cell line was used in the following assays. *Cd47*-/- Lin- BMCs, compared to WT Lin- BMCs, showed significantly less TEM through *Cd47*-/- MS1 cells (*Figure 4F*). Similarly, genetic deficiency of CD47 or blockade of CD47 signal on T lymphocytes also showed reduced TEM (*Figure 4—figure supplement 1K, L*). Thus, CD47 expression on immune cells is correlated with enhanced transmigration. These data suggest that elevated CD47 on migrating progenitors may engage SIRPα on specialized TPECs to promote their thymic entry.

## SIRPα intracellular signal controls TEM via SHP2 and Src

CD47-SIRPα interaction could transduce signaling bidirectionally. We next examined whether SIRPα signals through ECs for TEM regulation. To do this, an EC line lacking intracellular domain of SIRPα was constructed (*Inagaki et al., 2000*) (*Sirpa*-ΔICD MS1) (*Figure 5—figure supplement 1A*). *Sirpa*-ΔICD MS1 cells did not abolish surface display of the molecule and considerable expression of extracellular region of SIRPα is detected (*Figure 5—figure supplement 1B*), which would retain its ability to engage CD47 for potential signaling downstream of CD47 in interacting cells. However, T lymphocyte transmigration across *Sirpa*-ΔICD MS1 cells reduced to the level similar to that on *Sirpa*-/- MS1 cells (*Figure 5A*), suggesting SIRPα regulates TEM mainly through ECs. Furthermore, lentiviral transduction with WT *Sirpa* coding sequence (*Sirpa*-WT) in *Sirpa*-/- MS1 cells largely recovered TEM (*Figure 5B*), whereas transduction with loss-of-function mutant *Sirpa*-Y4F, in which all four functional tyrosine residues in the cytoplasmic ITIM motifs were substituted with inactive phenylalanine, failed

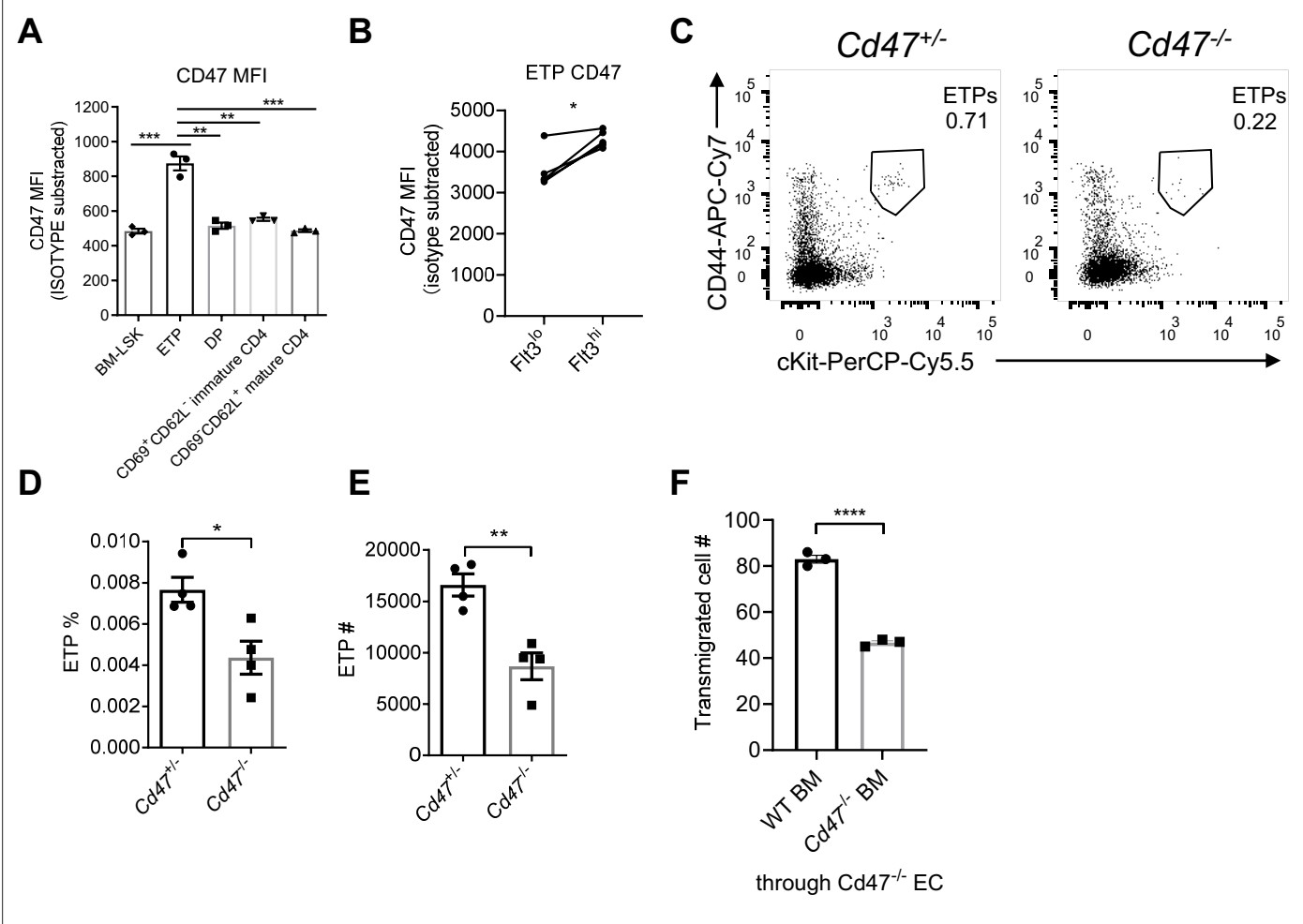

**Figure 4.** Migrating cell-derived CD47 guides their transendothelial migration (TEM). (**A**) CD47 expression measured by flow cytometry in various subsets of T cell lineage. BM-LSK: Lineage⁻Sca1⁺cKit⁺ lymphoid progenitor cells in bone marrows, n = 3 in each group, data are representative of three independent experiments. (**B**) Statistic analysis of CD47 expression on Flt3$^{hi}$ (newly immigrated) and Flot3$^{lo}$ (settled) early T cell progenitors (ETPs), gated as in *Figure 4—figure supplement 1B*, n = 3 in each group, paired t-test were performed, data are representative of three independent experiments. (**C–E**) Flow cytometric analysis (**C**), proportion (**D**), and corresponding cell numbers (**E**) of ETP population in *Cd47⁻/⁻* or control mice, n = 4 in each group, data are representative of three independent experiments. (**F**) Transmigration of Lin⁻ BMCs through *Cd47⁻/⁻* endothelial cells (ECs). FACS-sorted wild type (WT) or *Cd47⁻/⁻* Lin⁻ BMCs were driven by CCL19 (20 ng/ml) for 7 hr, n = 3 in each group, data are representative of three independent experiments. Error bars represent s.e.m. Asterisks mark statistically significant difference, *p < 0.05, **p < 0.01, ***p < 0.001, determined by two-tailed unpaired Student's t-test if not otherwise indicated. Source data are available in *Figure 4—source data 1*.

The online version of this article includes the following source data and figure supplement(s) for figure 4:

**Source data 1.** Source data file for Figure 4.

**Figure supplement 1.** Migrating cell-derived CD47 guides their transendothelial migration (TEM).

**Figure supplement 1—source data 1.** Source data file for Figure 4-figure supplement 1.

in TEM rescue (*Figure 5B*), even with restored surface expression (*Figure 5—figure supplement 1C*). Thus, SIRPα intracellular ITIMs-mediated downstream signaling in ECs is required for TEM.

Phosphorylation of ITIMs of SIRPα cytoplasmic tail may recruit and activate ubiquitously expressed tyrosine phosphatase SHP2 in ECs (*Takada et al., 1998*). To investigate whether SHP2 is sufficient to enable TEM in the absence of SIRPα signaling, *Sirpa⁻/⁻* MS1 cells were transduced with a lentiviral vector to express constitutively active SHP2 (E76K; *Bentires-Alj et al., 2004*, hereafter called *Shp2-CA*). *Shp2-CA* MS1 cells permitted significantly more TEM of T lymphocytes (*Figure 5C*), demonstrating that SHP2 activation can rescue the defective effect of SIRPα deficiency.

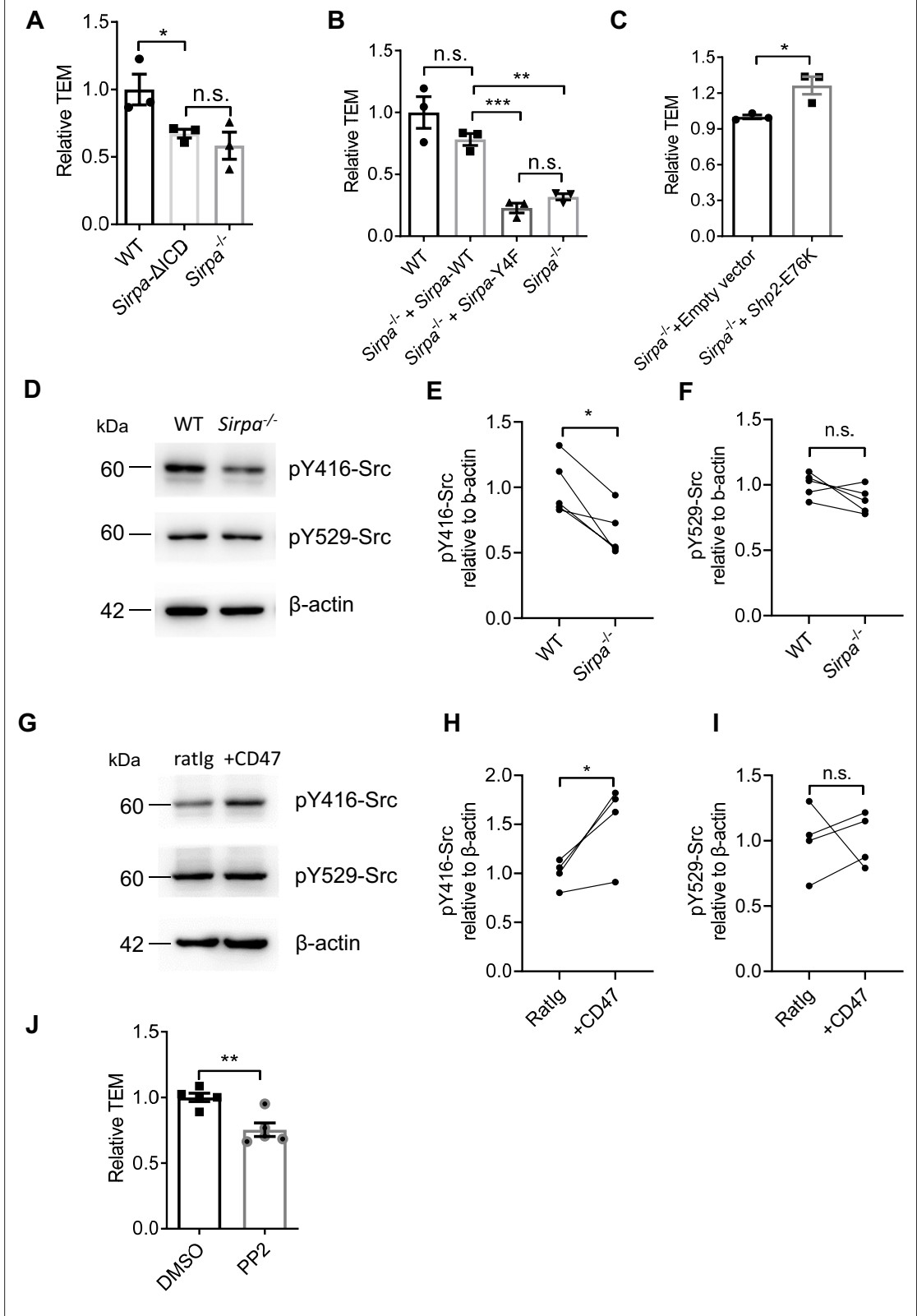

**Figure 5.** Signal regulatory protein alpha (SIRPα) intracellular signal controls transendothelial migration (TEM) via SHP2 and Src. (**A**) Relative TEM efficiency of lymphocytes through wild type (WT), intracellular-truncated *Sirpa*-ΔICD, and *Sirpa*⁻/⁻ MS1 endothelial monolayer, n = 3 in each group, data are representative of three independent experiments. (**B**) Relative TEM efficiency of lymphocytes through *Sirpa* overexpressing (*Sirpa*⁻/⁻ + *Sirpa* WT) or SIRPα-tyrosine-to-phenylalanine mutant overexpressing (*Sirpa*⁻/⁻ + *Sirpa*-4F) MS1 endothelial monolayer, n = 3 for each group, data are representative

*Figure 5 continued on next page*

*Figure 5 continued*

of three independent experiments. (**C**) Relative TEM efficiency of lymphocytes through *Sirpa*$^{-/-}$ MS1 endothelial monolayer overexpressing constitutively active form of SHP2 (+*Shp2* CA) or control empty vector (+Empty vector), n = 3 for each group, data are representative of three independent experiments. (**D**) Detection of Src activity in *Sirpa*$^{-/-}$ and WT MS1 endothelial cell lines, as measured by Western blot with anti-pY416-Src (active form Src antibody), anti-pY529-Src (inactive form Src antibody), and anti-β-actin control antibody. (**E,F**) Relative quantification of the active Src (pY416-Src) (**E**) and inactive Src (pY529-Src) (**F**) to β-actin in five tests, the relative expression in each sample is normalized to WT average, n = 4 in each group, data are representative of five independent experiments. Signifcance was analyzed by paired t-test. (**G**) Detection of Src activity in *Cd47*$^{-/-}$ MS1 cells upon mCD47-Ig or control Ig stimulation, as measured by Western blot. (**H,I**) Relative quantification of the active Src (pY416-Src) (**H**) and inactive Src (pY529-Src) (**I**) to β-actin in four tests, and the relative expression in each sample is normalized to WT average, n = 4 in each group, data are representative of four independent experiments. Significance was analyzed by paired t-test. (**J**) Lymphocyte transmigration, in the absence or presence of Src inhibitor PP2 (5 μM, 12 hr) through WT MS1 endothelial monolayer, n = 5 for each group, data are representative of three independent experiments. Error bars represent s.e.m. Asterisks mark statistically significant difference, *p < 0.05, **p < 0.01, ***p < 0.001, n.s., not significant, determined by two-tailed unpaired Student's t-test unless otherwise indicated. Raw images and statistics are available in *Figure 5—source data 1*.

The online version of this article includes the following source data and figure supplement(s) for figure 5:

**Source data 1.** Source data file for Figure 5.

**Figure supplement 1.** Construction of signal regulatory protein alpha (SIRPα) mutant MS1 cell lines.

**Figure supplement 1—source data 1.** Source data file for Figure 5-figure supplement 1.

---

Several studies have illustrated that SHP2 activity could cooperate activation of Src kinase in ECs (*Liu et al., 2012*; *Zhang et al., 2004*). We next investigated whether Src participates TEM regulation in the context of SIRPα signaling. In *Sirpa*$^{-/-}$ MS1 cells, active Src (pTyr-416) was significantly deficient compared to WT (*Figure 5D and E*) without affecting the pool of autoinhibitory inactive form of Src (pTyr-529) (*Figure 5D and F*). Furthermore, plate-bound CD47 significantly promoted Src activation (pTyr-416,) but not pTyr-529 in *Cd47*$^{-/-}$ MS1 cells (*Figure 5G–I*). These data suggest the ability of SIRPα downstream signaling in activating Src kinase.

To directly assess the role of Src in leukocyte transmigration, a specific inhibitor of Src family kinases, PP2, was applied to the WT MS1 cells during transwell assay. Significantly inhibited lymphocyte transmigration was observed upon PP2 treatment (*Figure 5J*). Thus, these data suggest that SIRPα downstream signaling activates SHP2 and Src kinase for the control of TEM.

## CD47-SIRPα signaling promotes VE-cadherin endocytosis

VE-cadherin is the dominant gate-keeping molecule controlling leukocyte transmigrating through endothelial barrier (*Corada et al., 1999*; *Vestweber, 2007*; *Vestweber et al., 2009*; *Wessel et al., 2014*). VE-cadherin controls junction opening by regulated destabilization and endocytosis from junctional area of cell surface (*Allingham et al., 2007*; *Benn et al., 2016*; *Wessel et al., 2014*). In addition, phosphatase SHP2 and kinase Src have been both reported to destabilize adherens junction during transmigration by promoting endocytosis of VE-cadherin (*Allingham et al., 2007*; *Wessel et al., 2014*). To test the role of endothelial SIRPα signal on VE-cadherin endocytosis, we adopted an in vitro VE-cadherin endocytosis assay as previously described (*Wessel et al., 2014*). The surface expression of VE-cadherin is not altered in *Sirpa*$^{-/-}$ ECs (*Figure 6—figure supplement 1A, B*). Upon T lymphocyte engagement, *Sirpa*$^{-/-}$ MS1 showed significantly lower T-cell-induced VE-cadherin endocytosis (*Figure 6A and B*). *Sirpa*-ΔICD MS1 cells showed similar unresponsiveness of VE-cadherin endocytosis to lymphocyte engagement (*Figure 6C and D*), demonstrating the requirement of SIRPα intracellular signaling in facilitating VE-cadherin endocytosis. Furthermore, inhibition of SIRPα downstream molecule Src by PP2 also significantly inhibited endocytosis of VE-cadherin in WT MS1 cells but not in *Sirpa*$^{-/-}$ MS1 cells (*Figure 6E*), suggesting an important role of Src activity downstream of SIRPα intracellular signaling in controlling VE-cadherin endocytosis.

VE-cadherin endocytosis is a quite dynamic process induced upon cellular contact. In an effort to analyze the effect of CD47-SIRPα signaling on this process, we set up an in vitro live imaging procedure to analyze VE-cadherin endocytosis at sites of migrating cell contact (*Kroon et al., 2014*). T lymphocytes were labeled with different fluorescent antibodies, treated with CD47-blocking reagent (CV1) or control hIg, mixed equally and then infused into a flow chamber mimicking physiological blood flow. The junctional retainment of VE-cadherin was measured by its colocalization with adhesion molecule CD31 (*Figure 6—figure supplement 1C*). VE-cadherin colocalization with CD31 under area of migrating cells was calculated and grouped by the different fluorescent labels on lymphocytes. The

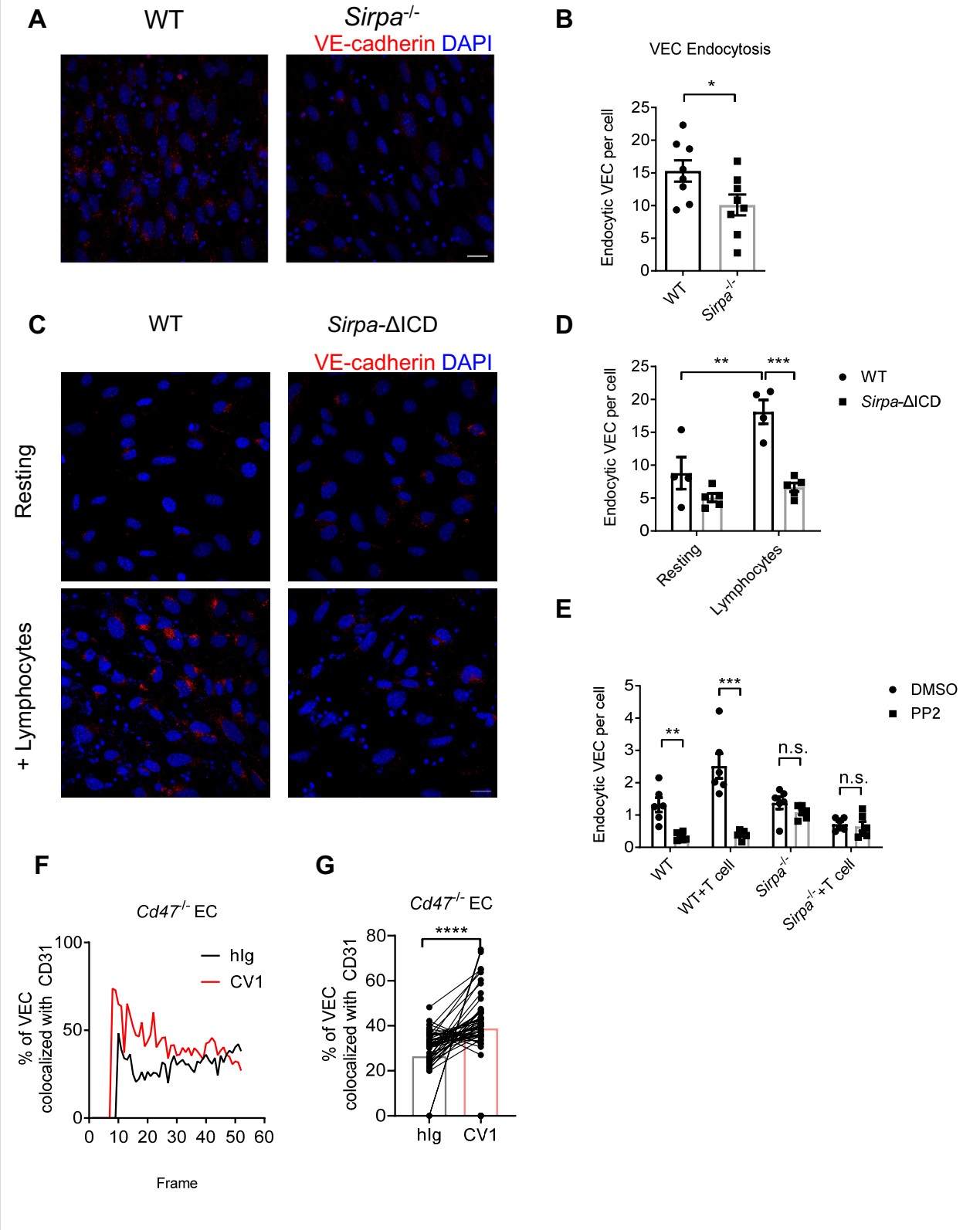

**Figure 6.** CD47-signal regulatory protein alpha (SIRPα) signaling promotes VE-cadherin endocytosis. (**A**) Representative imaging of endocytosed VE-cadherin in the presence of lymphocytes, scale bars represent 20 μm. (**B**) Statistical analysis of VE-cadherin endocytosis in MS1 endothelial cells. VE-cadherin fluorescence signal was quantified by ImageJ with same threshold for each slide. Data are representative of three independent experiments with n = 8 in each group (captured filed). (**C,D**) VE-cadherin endocytosis in intracellular-truncated SIRPα-ΔICD or wild type (WT) MS1 cells, with

*Figure 6 continued on next page*

*Figure 6 continued*

(+lymphocytes) or without lymphocyte incubation (resting). Representative confocal imaging of endocytosed VE-cadherin is shown (**C**) with statistical analysis of VE-cadherin endocytosis (**D**), n = 4 in WT, n = 5 in SIRPα-ΔICD-B2, data are representative of three independent experiments. (**E**) VE-cadherin endocytosis in the presence of lymphocytes (+T cell) and inhibitor of Src activation (+PP2, 5 μM, 2 hr). (**F,G**) Real-time analysis of adherens junctional VE-cadherin in the presence of migrating lymphocytes pretreated with CV1 or control hIg. Dynamic measurement (**F**) and statistical analysis (**G**) of VE-cadherin colocalization with CD31 after CD4⁺ T cell injection to the flow, data measurement began with emergence of T cells in the field and measured one time each frame for a total of 51 frames. Data were pooled in each group and analyzed by two-tailed paired t-test. Each pair of data indicates paired signals in the same frame. Data are representative of two independent experiments. Error bars represent s.e.m. Asterisks mark statistically significant difference, *p < 0.05, **p < 0.01, ***p < 0.001, n.s., not significant, determined by two-tailed unpaired Student's t-test unless otherwise indicated. Raw data and analysis method of endocytosis assay and real-time image statistics are available in *Figure 6—source data 1*.

The online version of this article includes the following source data and figure supplement(s) for figure 6:

**Source data 1.** Source data file for Figure 6.

**Figure supplement 1.** CD47-signal regulatory protein alpha (SIRPα) signaling promotes VE-cadherin endocytosis.

**Figure supplement 1—source data 1.** Source data file for Figure 6-figure supplement 1.

higher colocalization of VE-cadherin with CD31 indicates less VE-cadherin endocytosis. The test was performed on *Cd47⁻/⁻* MS1 cells to exclude potential influence of CD47 on ECs. Significantly higher degree of VE-cadherin/CD31 colocalization was found under migrating cells that were pretreated with CV-1 compared to that under control treatment (*Figure 6F and G*), suggesting the requirement of CD47 signal derived from migrating cells for induction of VE-cadherin endocytosis. Reciprocal fluorescence labeling of CV1- and hIg-treated migrating cells showed similar results (*Figure 6—figure supplement 1D, E*). These data suggest that upon migrating cell contacts, CD47-SIRPα signal regulates VE-cadherin endocytosis via activation of SHP2 and Src kinase, at sites of contact, to control TEM.

Taken together, we discovered SIRPα is specifically expressed by TPECs, and immigrating progenitor cell-derived CD47 initiates SIRPα intracellular signaling which induces VE-cadherin endocytosis for efficient transmigration of progenitor cells to the thymus for proper T cell development (*Figure 7*).

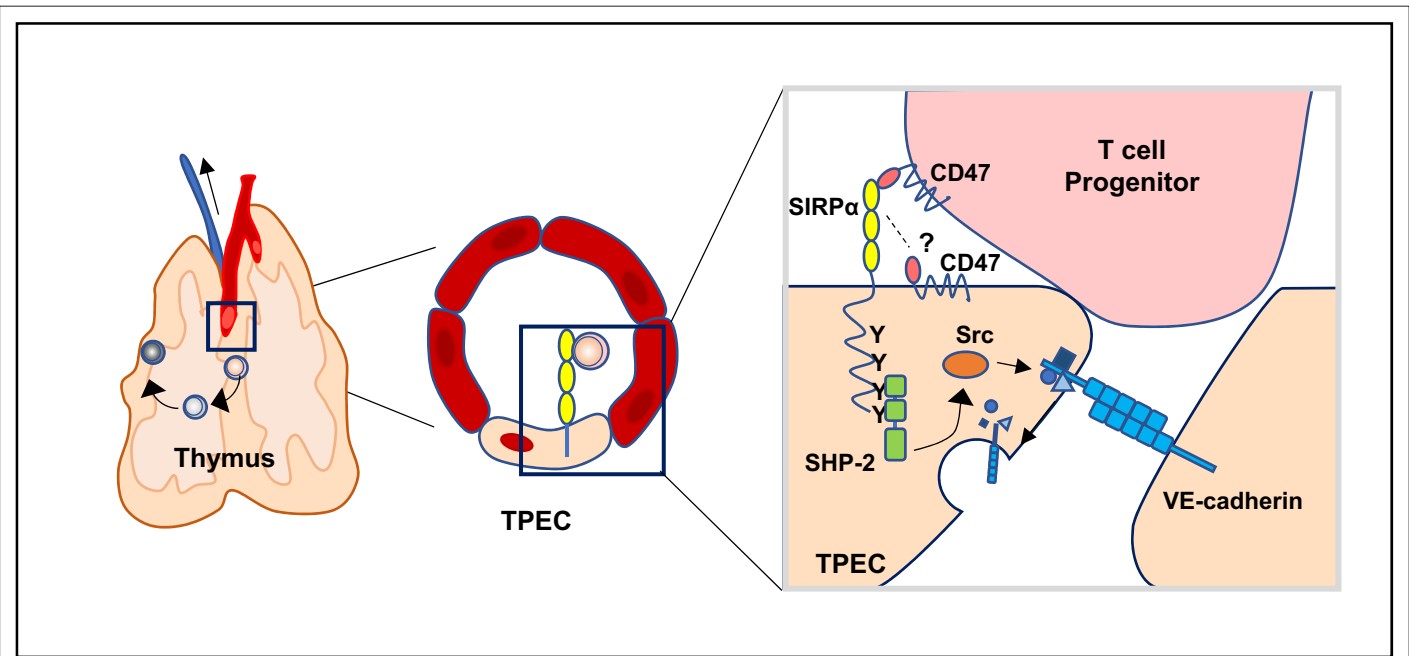

**Figure 7.** Schematic diagram of signal regulatory protein alpha (SIRPα) signaling in thymic portal endothelial cells (TPECs). SIRPα on TPECs is engaged by CD47 on progenitor cells to initiate an inward signaling which induces VE-cadherin endocytosis and transendothelial migration for thymic homing of the progenitors.

## Discussion

Homing of bone marrow-derived progenitors to the thymus is the prerequisite of continuous T cell development. In our previous study, Ly6C⁻P-selectin⁺ portal ECs, TPECs, were found to be the entry gate of HPCs to the thymus (*Shi et al., 2016*). In current study, we have further extended the knowledge about how TPECs interact with progenitors and regulate their thymic entry.

SIRPα has been well recognized as a 'don't eat me' receptor on myeloid cells. Interestingly, our current study reveals a previously totally unrecognized function of SIRPα on ECs for TEM regulation. SIRPα-CD47 interaction has been reported to regulate TEM of neutrophils and monocytes (*de Vries et al., 2002*; *Liu et al., 2002*; *Stefanidakis et al., 2008*). In these studies, CD47, but not SIRPα, is considered as the receptor-mediating signal transduction, while SIRPα expressed on neutrophils or monocytes is the ligand. SIRPα engagement of CD47 on epithelial cells or CEs activates Gi-protein-dependent pathway to facilitate transmigration or Rho family GTPase-dependent pathway to reorganize EC cytoskeleton. However, in our current work, we clearly showed that SIRPα downstream signal in ECs is required for VE-cadherin endocytosis and TEM. First, in bone marrow chimeric mice that were deficient in hematopoietic SIRPα, thymic ETP is normally maintained, while non-hematopoietic deficiency of SIRPα resulted in impaired thymic ETPs and thymic progenitor homing. These results exclude the possibility in the conventional view that SIRPα is derived from the migrating cells while CD47 is from endothelial compartment. Second, in the absence of functional intracellular region of SIRPα, while keeping extracellular region intact, *Sirpa*-ΔICD or *Sirpa*-Y4F EC monolayer showed deficiency of TEM comparable to that of *Sirpa*⁻/⁻ ECs, underlying the importance of EC-SIRPα downstream signal in controlling TEM. This was further confirmed by the fact that replenishment of WT *Sirpa* but not mutant *Sirpa*-Y4F can rescue TEM reduction in *Sirpa*⁻/⁻ ECs. Third, *Sirpa*-ΔICD ECs also had impaired endocytosis of VE-cadherin similar to that of *Sirpa*⁻/⁻ ECs, further supporting the role of EC-SIRPα downstream signaling in TEM.

The role of CD47 in this scenario might be complicated. CD47 is ubiquitously expressed in almost all cells, including immune cells and ECs both of which are involved in this trafficking process. Our data demonstrated that newly immigrated ETP expressed the highest level of CD47 than other subsets of hematopoietic cells during T cell development. The expression level of CD47 on ETPs is also significantly higher than that on TPECs. These data suggest that migrating cell-derived CD47 might play a major and active role in activating SIRPα on ECs for TEM. In fact, *Cd47*-deficient or blocked Lin⁻ BMCs or lymphocytes transmigrate across ECs less efficiently compared to WT control cells when endothelial CD47 is absent. Whether physiologically low level of endothelial CD47 constantly work for TEM remains intriguing. Although lymphocyte transmigrate through WT endothelial monolayer more efficiently than *Cd47*-deficient endothelial monolayer, given the artificial hyperexpression of CD47 on immortalized WT MS1 cells, this does not necessarily reflect the physiological role of endothelial CD47 on thymic homing. Therefore, a more physiological experimental model is required for further elucidation on the role of endothelial CD47 during thymic HPC homing.

Our study reveals a novel mechanism regulating adherens junction VE-cadherin endocytosis. Both SHP2 and Src kinase have been reported to regulate VE-cadherin endocytosis, via modification of VE-cadherin in different manners (*Allingham et al., 2007*; *Wessel et al., 2014*). However, how they are activated remains unclear. Here, our data suggest that CD47-SIRPα might be one of the upstream signals. In our study, deletion of SIRPα or its cytoplasmic domain results in significant deficiency of TEM and VE-cadherin endocytosis. Regeneration of constitutively activated SHP2 largely rescues TEM in *Sirpa*⁻/⁻ ECs, while regeneration of ITIM motif mutant SIRPα fails to do so. Together with the fact that ITIM motif activates cytosolic SHP2 phosphatase (*Motegi et al., 2003*; *Tsuda et al., 1998*), these data suggest that SIRPα signal-induced VE-cadherin endocytosis is at least in part through ITIM-SHP2 pathway. In addition, we have also found that CD47 engagement induces Src activation in ECs. Inhibition of Src kinase via PP2 results in significantly impaired TEM and VE-cadherin endocytosis in WT ECs, but not in *Sirpa*⁻/⁻ ECs. Thus, Src kinase is also involved in SIRPα signal-induced VE-cadherin endocytosis. It has been reported that SHP2 activation can regulate Src family kinase activity by controlling Csk recruitment (*Zhang et al., 2004*). Therefore, it is possible that CD47-SIRPα signal activates SHP2-Src axis to control VE-cadherin endocytosis and TEM. It should be noted that the EC cell line we used here is not derived from or corresponding to TPECs. More proper cell lines or conditional gene-deficient mice might be used for further confirmation of this signaling mechanism in

TPECs. On the other hand, whether this signaling mechanism is applicable in other settings is worthy of being investigated in future.

The defective thymic homing of progenitor cells in the absence of CD47-SIRPα signaling may lead to impaired thymic T cell development. In *Sirpa*-/- mice, at steady state, thymocyte development shows slight and significant defect at DN stage compared to that in WT mice, while later stage development appears normal. In another scenario, upon Lin- BMC adoptive transfer, thymocyte development in *Sirpa*-/- mice is also significantly impaired at both DN and DP stages. While no statistical significance was achieved, there is a trend of reduction of CD4+ and CD8+ SP thymocytes in *Sirpa*-/- mice. The biological consequence of the impaired thymic T cell development remains to be determined in various scenarios in future. Given the popular application of CD47-SIRPα blockade in tumor immunotherapy (*Feng et al., 2019*), its impact on T cell generation/regeneration and therapeutic efficacy might be of particular interest.

In summary, we have discovered a novel function of thymic endothelial SIRPα in controlling thymic progenitor homing and T cell development, and have revealed that migrating cell-derived CD47 and EC-SIRPα intracellular signal induce junctional VE-cadherin endocytosis and promote TEM. This study also provides novel insight on how this CD47- SIRPα should be manipulated for tumor immunotherapy.

## Materials and methods
### Mice
WT C57BL/6 mice were purchased from Vital River, a Charles River company in China; *Tek*cre mice were purchased from Nanjing Biomedical Research Institute (Nanjing, China). *Sirpa*flox/flox mice were kindly provided by Dr Hisashi Umemori (Children's Hospital, Harvard Medical School, Boston, MA); *Sirpa*-/- mice were kindly provided by Dr Hongliang Li (Wuhan University, China); *Cd47*-/- mice were gift from Dr Yong-guang Yang (The First Bethune Hospital of Jilin University, China). All mice are on C57BL/6 background and were maintained under specific pathogen-free condition. Mice for experiments are 4–8 weeks of age and sex-matched unless otherwise specified. This study was performed in strict accordance with approved protocols of the institutional committee of the Institute of Biophysics, Chinese Academy of Sciences, and all animal experimental procedures were performed in an effort to minimize mouse suffering.

### Cell lines
Mouse pancreatic islet EC line MS1 (CRL-2279) were purchased from American Type Culture Collection (ATCC). Mycoplasma contamination was tested negative. MS1 cells were cultured in DMEM (Hyclone) supplemented with 10% FBS (BI) and 1% penicillin-streptomycin (Gibco).

- *Sirpa*-/- and *Cd47*-/- MS1 were constructed by transfecting modified pLentiCRISPR v2 (addgene, Feng Zhang, #52961) cloned with oligos for desired single guide RNA. Two pairs of oligos for *Sirpa* targeting were designed for in case one of them fails to work.
  *Sirpa*-1-F: caccGCAGCGGCCCTAGGCGGCCA,
  *Sirpa*-1-R: aaacTGGCCGCCTAGGGCCGCTGC;
  *Sirpa*-2-F: caccGCCCGGCCCCTGGCCGCCTA
  *Sirpa*-2-R: aaacTAGGCGGCCAGGGGCCGGGC;
- MS1 cell line with truncated *Sirpa* lacking intracellular domain (*Sirpa*-ΔICD) was constructed by transfecting pLentiCRISPR v2 cloned with oligos targeting exon 6 of the first cytoplasmic region from N-terminus. The resulting clones were sequenced for ±1 frameshift, which created frameshift and advanced stop codon within exon 7. Sirpα ITIM motifs are within exon 8.
  *Sirpa*-ΔICD-E6-1-F: caccGAGGGGTCAACATCTTCCACA,
  *Sirpa*-ΔICD-E6-1-R: aaacTGTGGAAGATGTTGACCCCTC;
- *Sirpa*-4F and *Sirpa*-WT overexpressing lines were constructed based on *Sirpa*-/- MS1. Full-length Sirpα coding sequence was cloned from WT MS1 cells, and point mutated at all four tyrosine (Y) position from TAC or TAT to TTC or TTT, respectively, thus become non-functional phenylalanine (F). Cloning primers for *Sirpa* CDS:
  XbaI_*Sirpa*-F: ATATTCTAGACC ACCATGGAGCCCGCCGGCCCG,
  *Sirpa*_BamHI-R: TATGGATCCTCACTTCCTCTGGACCTGGA.
  WT or mutated (4F) form of *Sirpa* was ligated to pTK643-GFP with multiple cloning site and packaged for lentivirus. MS1 cells were infected for 24 hr with polybrene before seeded for

monoclonal selection. FACS analysis of GFP and SIRPα surface staining was used to identify overexpressed clones.

- *Shp2*-E76K overexpressing line was constructed based on *Sirpa*$^{-/-}$ MS1. Full-length Shp2 coding sequence was cloned from mouse genome and point mutated at glutamic acid 76 (E76) to constitutively active lysine (K). *Shp2*-E76K was then ligated to pTK643-GFP and *Sirpa*$^{-/-}$ MS1 was infected as above described. Cloning primer for *Shp2* CDS:

  XbaI-*Shp2*-F: ATTtctagaGCCACCatgACATCGCGGAGATGG,
  XbaI-*Shp2*-R: cgtctagaaTCATCTGAAACTCCTC TGCT

## Isolation of thymic ECs

Thymus was collected, digested, and Percoll enriched as previously described (*Shi et al., 2016*). Briefly, thymus was digested in RPMI 1640 medium with 2% fetal bovine serum, 0.2 mg/ml collagenase I, 1 U/ml dispase, and 62.5 µg/ml DNase for four rounds of 20 min digestion on a 37°C, 120 rpm shaker. Dissociated cells after each round and the final digest were washed and applied for discontinuous Percoll gradient enrichment. Cells were resuspended with 1.115 g/ml Percoll at the bottom, and 1.065 g/ml Percoll and PBS were laid on the middle and top layer, respectively. After being centrifuged at 2700 rpm (872 *g*) at 4°C for 30 min, EC-containing cells from the upper interface were collected and subjected to subsequent analysis.

## Cell preparation

Bone marrow cells were isolated from mouse femurs and tibias. In brief, soft tissues were cleared off and the ends of the bones were cut, bone marrow was than flushed out using a 23-gauge needle containing ice-cold PBS. Whole-tissue suspensions of thymus, lymph node, and spleen were generated by gently forcing the tissue through a 70 µm cell strainer. Red blood cells were lysed with ACK in samples of bone marrow and spleen.

## Flow cytometry and cell sorting

Flow cytometry data were acquired with an LSRFortessa cell analyzer (BD Biosciences) and analyzed using FlowJo software (BD Biosciences). Cell sorting was performed on a FACSAria III cell sorter (BD Biosciences). The following fluorescent dye-conjugated antibodies against cellular antigens were used for: (1) analysis of the thymic portal endothelia cells (TPECs): CD45 (30-F11), CD31 (MEC13.3), P-selectin (RB40.34), and Ly-6C (HK1.4); SIRPα (P84) (2) analysis of the lineage-negative progenitor cells: CD45.1 (A20), CD45.2 (104), lineage cocktail: CD11b (M1/70), CD11c (N418), NK1.1 (PK136), Gr-1 (RB6-8C5), B220 (RA3-6B2), and TER-119 (Ter-119); (3) analysis of the ETPs: lineage cocktail (see above), CD4 (GK1.5), CD8 (53-6.7), CD25 (PC61), CD44 (IM7), and c-Kit (2B8); (4) analysis of LSK and CLPs: lineage cocktail (see above), c-Kit (2B8), Sca-1 (D7), IL7Rα (A7R34), and Flt3 (A2F10); (5) analysis of peripheral lymphoid subsets: CD4 (GK1.5), CD8 (53-6.7), B220 (RA3-6B2), CD62L (MEL-14), CD44 (IM7). (6) Other antibodies used: CD47 (miap301), VE-cadherin (BV13), VCAM-1 (429). (7) Corresponding isotypes: Rat IgG1 κ iso (RTK2071), Rat IgG2a κ iso (eBR2a). Dead cells were excluded through DAPI (Sigma-Aldrich) or LIVE/DEAD (Thermo Fisher) positive staining.

## Quantitative real-time PCR

Thymic ECs were gated as subset I (P-selectin$^-$Ly6C$^+$), subset II (P-selectin$^+$Ly6C$^+$), and subset III, which is TPEC (P-selectin$^+$Ly6C$^-$). These three subsets were FACS-sorted and RNA was extracted using RNeasy Micro Kit (Qiagen). The quality and quantity of total RNA was assessed using a Nanodrop 2000c spectrometer (Thermo Scientific). cDNA was synthesized using RevertAid First Stand cNDA Synthesis Kit (Thermo) and Oligo (dT)$_{18}$ primer. Gene expression was quantified using the following primers. Quantitative real-time PCR was performed using SYMBR Premix Ex Taq (Takara) and run on Applied Biosystems 7500 Real-Time PCR System. Relative mRNA expression was calculated with the $2^{-\Delta\Delta CT}$ method.

*Sirpa*-F: TGCTACCCACAACTGGAATG,
*Sirpa*-R: CCCTTGGCTTTCTTCTGTTT,
*Actb*-F: ACACCCGCCACCAGTTCGC,
*Actb*-R: ATGGGGTACTTCAGGGTCAGG;
*Gapdh*-F: AACCACGAGAAATATGACAACTCACT,

*Gapdh*-R: GGCATGGACTGTGGTCATGA.

For chemokine detection on TPEC, Percoll-enriched thymic stromal cells were FACS-sorted for TPECs and extracted for RNA, and technical replicate was merged.

*Ccl19*-F: GTGGATCGCATCATCCGAAG
*Ccl19*-R: CTCAAGACACAGGGCTCCTT
*Ccl21a*-F: GGGTCAGGACTGCTGCCTTAAG
*Ccl21a*-R: AGCTCAGGCTTAGAGTGCTTCC
*Ccl25*-F: GCTTTTTGCCTGCCTGGTTG
*Ccl25*-R: TCAGTCTGAGAGTCTGAGGC

## SIRPα-hIg (CV1) production

SIRPα-hIg was produced as previously described (*Liu et al., 2015*). Briefly, pEE12.4-CV1, kindly provided by Dr Yang-Xin Fu (University of Texas Southwestern Medical Center, Dallas, TX), was transiently expressed in FreeStyle 293 expression system (Thermo Fisher). SIRPα-hIg was then purified with Sepharose Protein A/G beads. In in vitro TEM assay, lymphocytes were incubated with 10 µg/ml CV1 in PBS at 4°C for 15 min and washed with culture medium before applied to the transwell.

## Bone marrow chimeras and adoptive transfer of total bone marrow or Lin- BMCs

Six-week-old recipient mice received 10 Gy γ radiation once. $5 \times 10^6$ donor bone marrow cells were injected intravenously into the recipients on the day or the next day of irradiation. Eight weeks later, mice are ready for subsequent homing studies. Short-term homing assay: donor bone marrows were taken from femurs and tibias from WT mice, red blood cells were lysed and labeled by 2 µM CFSE, and $5 \times 10^7$ bone marrow cells were transferred intravenously to the recipients. The recipients were sacrificed 2 days later and detected for donor-derived progenitors in the thymus and spleen. Lin- BMC short-term homing and reconstitution: BM cells were taken as previously described, and labeled by lineage markers with PE conjugates (CD11b, CD11c, Ly6C, Ter119, B220, CD4, CD8, NK1.1), a secondary biotin anti-PE antibody was applied, before labeled with Streptavidin-MACS magnetic beads, finally cells were applied to MACS LD cell separation column (Miltenyi), and the flow-through cells were collected as Lin- BMC-enriched cells.

## Short-term homing and development of lineage- progenitors

CD45.1+ Lin- BMCs were enriched by MACS-negative selection, following FACS sorting lineage- proportion. $1 \times 10^6$ Lin- BMCs were i.v. injected into CD45.2+ recipients of WT or *Sirpa*-/- mice. CD45.1+ donor cells present in the thymus were detected 48 hr later for short-term homing, or 3 weeks later for T cell development.

## Lymphocyte adhesion assay

The adhesion assay was performed mainly as previously described (*Lowe and Raj, 2015*). $1 \times 10^5$ /ml MS1 cells were plated on coverslip (NEST, $\phi$ = 15 mm), growing for 24 hr and then stimulated with 10 ng/ml TNFα for additional 24 hr. $1 \times 10^6$ total lymphocytes from axillary and inguinal lymph nodes were added to the top of coverslip and incubated for 3 hr. Dunk in and out the coverslip vertically of the PBS five times before trypsinized and collect total cells for lymphocyte counting by FACS.

## Transmigration assay

$2.5 \times 10^4$ MS1 cells were initially plated on transwell filter (Corning, $\phi$ = 6.5 mm, with 5.0 µm pore), allowed growing for 24 hr and then stimulated with 10 ng/ml TNFα for additional 24 hr before transmigration assay. $5 \times 10^4$ sorted Lin- BMCs or $1 \times 10^6$ total lymphocytes from inguinal lymph nodes were added per transwell in 100 µl complete DMEM culture medium. Chemoattraction was achieved by adding 10 ng/ml (for lymphocyte) or 20 ng/ml (for Lin- BMCs) CCL19 (PeproTech) in the bottom chamber in 600 µl complete DMEM culture medium. Seven hours (for Lin- BMCs) or 4 hr (for lymphocytes) or later, migrated cells were collected for cell counting and subset analysis by FACS.

For CV1 incubation before transmigration, lymphocytes were incubated with CV1 (10 µg/ml) or control hIg (10 µg/ml) for 30 min at 4°C before applied to the transwell.

## Confocal assay

Thymic cryosection were cut at the thickness of 8 μm. Sections were fixed in ice-cold acetone for 10 min and rehydrated with PBS for 5 min before antibody incubation. Ly6C-FITC, anti-Collagen IV(Rabbit pAb), CD31-AF647, and SIRPa-biotin were primarily applied overnight at 4°C and Alexa Fluor 555-goat anti-rabbit IgG, Sreptavidin-V450 were secondarily applied for 1 hr at room temperature. Slides were mounted with anti-fade medium and analyzed with Nikon A1R+ confocal microscopy. Images were captured with 40× objective at 1/24 frame per second with a resolution of 4096*4096 and 2× average setting.

## VE-cadherin endocytosis assay

The endocytosis assay was performed mainly as previously described (*Wessel et al., 2014*). In this assay, $1 \times 10^5$ /ml MS1 cells were plated on coverslip, allowed growing for 24 hr and then preactivated with 10 ng/ml TNFα for additional 24 hr. Cells grown to confluency were treated for 1 hr with 150 μM chloroquine prior to the endocytosis assay. Cell were then incubated for 30 min at 37°C with anti-VE-cadherin-biotin (BV13) in culture medium. Antibody was then washed and $1 \times 10^6$ total lymphocytes were added to the top of coverslip. After 1 hr incubation at 37°C, lymphocytes were removed by quickly rinsing wells with prewarmed culture medium. And surface-bound antibodies were removed by washed for three times, 20 s each time, with acidic PBS (pH 2.7 PBS with 25 mM glycine and 2% FBS). Cells were then fixed with 1% paraformaldehyde in PBS for 5 min followed by permeabilization with 0.5% Triton X-100 for 10 min at room temperature. Internalized primary antibodies were then detected by fluorescence-conjugated Streptavidin (Biolegend). DNA was stained with DAPI. Five or more fields were observed in each sample on a Zeiss LSM 700 confocal system (63×/1.4 Oil) with locked parameters. Endocytosed VE-cadherin was quantified via home-made script running under ImageJ batch mode with fixed signal threshold for each experiment, the final results were presented as arbitrary intracellular VE-cadherin signal per MS1 cell, average signal of all cells was calculated for each filed. Threshold value was adjusted according to signal background from isotype control sample.

## Western blotting

MS1 cells were lysed with lysis buffer containing 20 mM Tris·HCl, 2 mM EDTA, 0.5% NP-40, 1 mM NaF and 1 mM $Na_3VO_4$, and protease inhibitor cocktail. The samples were heated to 95° for 5 min with loading buffer containing 0.5% 2-mercaptoethanol. Equal amount of samples was loaded and resolved on a 10% SDS-PAGE gel. Proteins were then transferred to PVDF membrane (Millipore, 0.45 μm). The membranes were first incubated with anti-phospho-Src (Tyr416) antibody (CST) or anti-phospho-Src (Tyr529) antibody (Abcam) followed by Goat anti-Rabbit HRP (CWBIO, China). The blot was developed by chemiluminescent HRP substrate (ECL, Millipore). The blot was then stripped and reblotted with anti-β-actin antibody (Zsbio, China) subsequently. The images were captured on a Tanon 5200 chemi-image system (Tanon, China). Gel images were quantified with Lane 1D analysis software (SageCreation, China).

## VE-cadherin real-time imaging

Flow chamber was designed and made by Center for Biological Imaging (CBI), Institute of Biophysics (IBP), Chinese Academy of Sciences (CAS). Briefly, a coverslip (NEST, $\phi$ = 25 mm) with endothelial monolayer can be inserted into the thermostatic (37°C) flow chamber and supplied with prewarmed flow medium (DMEM). Flow medium was controlled by a pump (adjusted by frequency and voltage). The flow chamber was then fixed to an adaptor which allowed the bottom side of the coverslip fit into observation range of the 60× immersion oil lens of an Olympus FV1200 spectral inverted laser scanning confocal microscopy. MS1 ECs were plated on the coverslip sited in six-well plate at a density of $1.2 \times 10^5$/ml and activated by 10 ng/ml of TNFα 24 hr later. After additional 24 hr, MS1 ECs were sequentially labeled with rat-anti-mouse VE-cadherin, anti-Rat-Alexa Fluor 488 and anti-CD31-Alexa Fluor 647. CD4+ T cells were prepared from mouse inguinal and axillary lymph nodes, and divided into two parts: one part blocked by CD47 antagonist CV1 at 10 μg/ml and labeled by anti-CD4-V450, another part controlled by hIg and labeled by anti-CD4-PE. Two parts of CD4+ T cells were equally mixed and resuspend to $1 \times 10^6$/ml in flow medium. Focus was limited to the layer that maximized junctional CD31 signal, and Z-axis drift compensation was activated to lock focus. Continuous multi-channel imaging on AF488, PE, AF647, and V450 was conducted and concatenated. Colocalization of

VE-cadherin with CD31 at sites of differentially treated T cells was measured and calculated by Imaris 9 software. In reciprocally labeling tests (*Figure 6—figure supplement 1D, E*), CV1-treated T cells were labeled with anti-CD4-PE, and hIg-treated T cells were labeled with anti-CD4-V450.

## RNA-Seq and microarray data analysis

The RNA-Seq data of thymic ECs was previously published and available as GSE_83114 (*Shi et al., 2016*). Differentially expressed genes of TPECs (fold change >2 and p value of pairwise t-test <0.01 in either the contrast of TPECs versus subset II or subset I EC subset) were filtered for their function by gene ontology term cell migration (GO_0016477). The hints were candidates for further analysis. Microarray data from previously published work (*Lee et al., 2014*) were analyzed by Affymetrix Expression Console (1.4.1) and Transcriptome Analysis Console (3.0) software following manufacturer's instruction. Z-score normalized heatmap was generated by gplots package in R (3.6).

## Statistical analysis

Statistical analyses were performed using GraphPad 6.0 (Prism). Two-tailed paired or unpaired Student's t-test was used for significance test in unpaired data analysis. Two tailed paired Student's t-test was used for significance test in paired data analysis, as those in *Figure 5E, F, H,I*, *Figure 6G*, and *Figure 6—figure supplement 1E*. Error bars in figures indicate s.e.m. (n.s., not significant; *p < 0.05; **p < 0.01; ***p < 0.001; and ****p < 0.0001).

## Acknowledgements

We would like to thank Dr Hisashi Umemori (Children's Hospital, Harvard Medical School, Boston, MA, USA) for providing *Sirpa*<sup>flox/flox</sup> mice, Dr Hong-liang Li (Wuhan University, China) for *Sirpa*<sup>-/-</sup> mice, Dr Yong-guang Yang (The First Bethune Hospital of Jilin University, China) for *Cd47*<sup>-/-</sup> mice, and Dr Yang-Xin Fu (University of Texas Southwestern Medical Center, Dallas, TX, USA) for pEE12.4-CV1 expression plasmid. We would like to thank Xiaoyan Wang, and Yihui Xu in the Key Laboratory of Infection and Immunity, Institute of Biophysics (IBP), Chinese Academy of Sciences (CAS) for providing instrumental support on flow cytometry and confocal imaging. We would like to thank Yan Teng, Yun Feng, and Chunliu Liu from Center for Biological Imaging (CBI), Institute of Biophysics, Chinese Academy of Science for their help of perfusion fluorescence imaging and confocal images analysis, their innovative flow chamber system made the real-time imaging of VE-cadherin in live EC lines possible. This work was supported by grants from National Natural Science Foundation of China (31770959 and 82025015 to MZ).

## Additional information

### Funding

| Funder | Grant reference number | Author |
| --- | --- | --- |
| National Natural Science Foundation of China | 31770959 | Mingzhao Zhu |
| National Natural Science Foundation of China | 82025015 | Mingzhao Zhu |

The funders had no role in study design, data collection and interpretation, or the decision to submit the work for publication.

### Author contributions

Boyang Ren, Data curation, Formal analysis, Investigation, Methodology, Validation, Visualization, Writing – original draft, Writing – review and editing; Huan Xia, Yijun Liao, Hang Zhou, Zhongnan Wang, Investigation; Yaoyao Shi, Investigation, Resources; Mingzhao Zhu, Conceptualization, Data curation, Formal analysis, Funding acquisition, Investigation, Methodology, Project administration, Resources, Software, Supervision, Validation, Writing – review and editing

## Author ORCIDs
Boyang Ren (iD) http://orcid.org/0000-0002-4157-5199
Mingzhao Zhu (iD) http://orcid.org/0000-0003-2001-2669

## Ethics
This study was performed in strict accordance to approved protocols of the institutional committee of the Institute of Biophysics, Chinese Academy of Sciences, and all animal experimental procedures were performed in an effort to minimize mouse suffering.

## Decision letter and Author response
Decision letter https://doi.org/10.7554/eLife.69219.sa1
Author response https://doi.org/10.7554/eLife.69219.sa2

## Additional files

### Supplementary files
• Transparent reporting form
• Supplementary file 1. Source data file for GSE_83114 RNA-seq analysis.

### Data availability
All data generated or analyzed during this study are included in the manuscript and supporting files.

The following previously published dataset was used:

| Author(s) | Year | Dataset title | Dataset URL | Database and Identifier |
|---|---|---|---|---|
| Yaoyao S, Weiwei W | 2016 | LTβR controls thymic portal endothelial cells for hematopoietic progenitor cell homing and T cell regeneration | https://www.ncbi.nlm.nih.gov/geo/query/acc.cgi?acc=GSE83114 | NCBI Gene Expression Omnibus, GSE83114 |

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
