## [Editor Report]

The primary audience that will be keenly interested in these findings will be those with an interest in T cell development and thymic function, however given the broad applicability of transendothelial migration, there will also likely be broader interest in these findings. The work provides key new insights as to the role of the CD47-SIRPa signaling axis in the regulation of how hematopoietic progenitors enter into the thymus to initiate T cell development.

---

## [Decision Letter]

**Decision letter after peer review:**

Thank you for submitting your article "Endothelial SIRPα signaling controls thymic progenitor homing for T cell regeneration and antitumor immunity" for consideration by *eLife*. Your article has been reviewed by 3 peer reviewers, including Juan Carlos Zúñiga-Pflücker as Reviewing Editor and Reviewer #1, and the evaluation has been overseen by Richard White as the Senior Editor. The following individual involved in review of your submission has agreed to reveal their identity: Jarrod A. Dudakov (Reviewer #3).

Overall, there was enthusiasm for the paper but some significant concerns remain. Below, I am listing what we consider essential revisions, but you can see the detailed reviewer comments below that so this can act as a guide for your experiments.

Essential revisions:

1) There was agreement by the reviewers that the results from the thymus reconstitution assay shown in Figure 7 require further analysis and controls for the conclusions to be supported. One recommendation is to remove this experimental approach and pursue it as part of a more complete study on the effects of CD47-SIRPa inhibition in immunotherapy.

2) The assays used to address thymus homing are not sufficiently strong to support the conclusions, and the use of a competitive mix bone marrow chimera would be more appropriate. As well as examining for Flt3+ ETPs as a better way to enumerate recently arrived ETPs.

3) It would be important for the authors to discuss or provide new insights as to the regulation of SIRPa expression by TPECs, is this regulated by ETP niche occupancy, LT signals, or some other mechanism.

4) Carefully check figure legends for reporting of individual mice and independent experiments for each panel, which will allow for a more rigorous reporting and data analysis.

*Reviewer #1 (Recommendations for the authors):*

The work presented by the authors convincingly illustrates the role played by SIRPa in TPECs to facilitate TSP entry into the thymus.

1 – Given the important role played by chemokines in attracting TSPs to the thymus, it would be important to show whether absence of SIRPa affects the expression of key chemokines by TPECs, which could further explain the observed decrease in thymic recruitment. This was shown for VCAM1 (Figure 3E), which nicely excludes this other potential explanation.

*Reviewer #2 (Recommendations for the authors):*

The authors should carefully check figure legends for reporting of individual mice and independent experiments for each panel. Frequently, 3 independent experiments with 3 mice each are reported and fewer dots for individual mice are shown. For instance, in figure 2, in some instances 4 dots per group are shown, which is more than an individual "representative" experiment but too few to be justified by elimination of outliers, as this would indicate more outliers than usable data points. The authors should check whether comparison of multiple groups requires ANOVA, such as in Figure 4A.

The authors report higher levels of CD47 on ETPs, when compared to bone-marrow progenitors and later stages. However, ETPs are already in the thymus, whereas the progenitors should be expected to have higher levels. The authors should reassess their interpretation of data. They could also subdivide ETPs according to Flt3 expression into young and old ETPs. CD47 flow cytometry only shows a minor shift over isotype even for ETPs (4E figure supplement 1). The authors should elaborate on their normalization procedure for cytometry data and also show primary data examples for cell types analyzed in Figure 4A.

For the experiments og thymus regeneration, it would strengthen the paper, if the authors performed additional homing assays such as those performed by Zlotoff et al., (2011) or used the conditional Tie2Cre-SIRPa-KO model.

The manuscript should be carefully checked for grammatical errors.

*Reviewer #3 (Recommendations for the authors):*

Overall this was a very well-conceived study with some important and novel findings. Although I do believe that the experiments detailed support the main conclusions of the manuscript, the following are a few questions/comments that would strengthen the manuscript if addressed.

– The authors have elegantly shown the importance of TPECs in guiding the entry of HPCs into the thymus, and in their previous work the authors have shown that TPECs associate with perivascular spaces. However, given that not all TPECs express Sirpa (Figure 1E), and in their previous work there are at least 20% of TPECs that do not associate with PVS (and it is not at all clear from their previous work what the proportion of TPECs are located at the CMJ where we expect HPCs to enter the thymus), it would be extremely beneficial to show the location of TPECs and Sirpa in the thymus in relation to other architectural features (such as PVS, CMJ etc.)

– In Figure 2 I appreciate the inclusion of BM progenitor subsets but it would be great if additional subsets could be retroactively included such as LMPPs. This is not critical but would strengthen the claims if possible.

– In Figure 7, is there any change in number of TOPECs in mice treated with CV1?

– As a stylistic note, specific figures/panels should not be called out in the discussion.

---

## [Author Response]

Essential revisions:1) There was agreement by the reviewers that the results from the thymus reconstitution assay shown in Figure 7 require further analysis and controls for the conclusions to be supported. One recommendation is to remove this experimental approach and pursue it as part of a more complete study on the effects of CD47-SIRPa inhibition in immunotherapy.

Thank you very much for taking time to review the manuscript. We agree further study on the effects of SIRP_α_ inhibition in immunotherapy is necessary. As CV1 inhibition arouse many other questions regarding immune checkpoint blockade, current approaches cannot thoroughly evaluate this effect and illuminate the mechanism behind. Therefore, these data were removed from this study.

2) The assays used to address thymus homing are not sufficiently strong to support the conclusions, and the use of a competitive mix bone marrow chimera would be more appropriate. As well as examining for Flt3+ ETPs as a better way to enumerate recently arrived ETPs.

We agree with these two points, and we have strengthened our conclusion on thymus homing by the following three aspects:

a) In the thymic short-term homing assay, we replaced total bone marrow cells with lineage^-^ progenitor cells, therefore significantly fewer cells (1×10^6^) were needed. This made the approach more physiologically relevant, more straightforward and less interfered by other components in the total bone marrows.

b) We tried to construct mixed bone marrow chimeras. In lethally irradiated WT (CD45.2) mice, 1:1 WT (CD45.2) and *Cd47^-/-^* total bone marrow cells were transferred i.v., 6 weeks later, chimerism of T cells in the peripheral blood were detected by FACS. To our surprise, we found both CD4^+^ and CD8^+^ T cells in the periphery are CD45.2^+^, indicating a significant defect of reconstitution of *Cd47^/-^* bone marrow cells (Author response image 1).

**Author response image 1. sa2fig1:** Reconstitution of mixed WT and *Cd47^-/-^* bone marrow chimeras. (A) Schematic view of mixed bone marrow chimera. (B) FACS analysis of chimerism of T cells in the peripheral blood in the host 6 weeks after bone marrow adoptive transfer. (C,D) Statistics of WT and *Cd47^-/-^* donor derived CD4^+^ (C) and CD8^+^ (D) T cells. n=10 in each group, paired t-test applied. ***: p <0.001.

To further examine the reconstitution defect of *Cd47^-/-^* bone marrow cells, we also tried mixed bone marrow progenitor cell transfer in non-irradiated WT mice. This time, we tested the development of donor progenitors in the thymus, as well as in the bone marrows for control of non-thymic factors. Significantly impaired development of T cells from *Cd47^-/-^* progenitor cells was similarly found (Author response image 2). In addition, similar degree of deficiency of *Cd47^-/-^* derived cells was also found in the bone marrows (Author response image 2), indicating the defect is probably at the progenitor cell stage. This is likely due to the phagocytotic clearance of CD47 deficient cells by phagocytes (Jaiswal et al., 2009). Therefore, it currently remains a technical difficulty to address the role of CD47 on progenitor cells in thymic homing using mixed competitive bone marrow chimeras, or mixed progenitor cell adoptive transfer in non-irradiated hosts.

**Author response image 2. sa2fig2:** Development of WT and *Cd47^-/-^* Lin^-^ progenitors in non-irradiated hosts. (A) Schematic view of the experiment. (B) Proportion of donor cells in the mixture before transfer. (C) Proportion of donor-derived thymocytes 2 weeks later. (D,E) Statistics of proportion of WT and *Cd47*^-/-^ donor derived cells in the thymus (D) and in the bone marrow (E) of the host. (F) Cell number of donor-derived thymocytes in each thymocyte subset. n=3 in each group, paired t-test applied. ***: p <0.001, ****:p<0.0001.

c) To avoid the complicated scenario in vivo as discussed above, we have used a clean in vitro transwell assay and have demonstrated that CD47 on the progenitors was indeed required for transmigration (new Figure 4F). In addition, comparison of Flt3^+^ newly immigrated ETPs and Flt3^-^ ETPs found higher CD47 expression on Flt3^+^ ETPs, indicating the involvement of CD47-SIRP_α_ signal in progenitor homing (new Figure 4B).

3) It would be important for the authors to discuss or provide new insights as to the regulation of SIRPa expression by TPECs, is this regulated by ETP niche occupancy, LT signals, or some other mechanism.

It’s an interesting and important question how SIRP_α_ expression is regulated on TPECs. Given the important role of LT-LT_β_R signaling on TPEC development and maintenance, we first tested whether LT-LT_β_R signal would be required for SIRP_α_ expression.

However, the remaining TPECs in *Ltbr^-/-^* mice showed similar level of SIRP_α_ expression compared to that in WT mice (new Figure 1—figure supplement 1C). Thymic stromal niche is another factor regulating thymic settling of progenitor cells (Krueger, 2018; Prockop and Petrie, 2004). Increased thymic stromal niche was found during irradiation (Zlotoff et al., 2011). We detected SIRP_α_ expression on TEPCs at Day 14 after 5.5Gy total body sublethal irradiation and found no significant change in SIRP_α_ expression upon irradiation (new Figure 1—figure supplement 1D). Whether SIRP_α_ expression on TPECs is a constitutive event or regulatable upon thymic microenvironmental change remains to be tested in future.

4) Carefully check figure legends for reporting of individual mice and independent experiments for each panel, which will allow for a more rigorous reporting and data analysis.

We apologize for the lack of clarity with these legends. We have now checked all legends and added specific description for each experiment.

We would like to thank the referees again for taking the time to review our manuscript.

Reviewer #1 (Recommendations for the authors):The work presented by the authors convincingly illustrates the role played by SIRPa in TPECs to facilitate TSP entry into the thymus.1 – Given the important role played by chemokines in attracting TSPs to the thymus, it would be important to show whether absence of SIRPa affects the expression of key chemokines by TPECs, which could further explain the observed decrease in thymic recruitment. This was shown for VCAM1 (Figure 3E), which nicely excludes this other potential explanation.

Thank you for your advice. In mice, chemokine receptor 7 (CCR7) and 9 (CCR9), especially the latter, play important roles for thymic homing of hematopoietic progenitors. Thymic ETP population was significantly reduced in CCR9 KO mice (about 7-fold) compared to WT mice at resting stage, while CCR7 KO mice show largely unchanged or even increased population of ETPs. More dramatic reduction of ETP population was found in CCR9 and CCR7 double KO mice (Krueger et al., 2010; Zlotoff et al., 2010). The corresponding chemokines, CCL25 and CCL19/CCL21, are expressed by both thymic ECs and thymic epithelial cells. To determine whether SIRP_α_ may regulate the expression of these chemokines, TPECs were sorted from wild type or *Sirp_α_*^-/-^ mice for quantitative RT-PCR assay. No change of *Ccl25* and *Ccl19* expression were found upon deficiency of Sirp_α_, and *Ccl21a* showed slight reduction. Considering unreduced thymic ETP population in CCR7 deficient mice, the reduced *Ccl21a* expression in *Sirp_α_*^-/-^ EC unlikely explains the significant ETP reduction in *Sirp_α_*^-/-^ mice. These results have been updated as part of the manuscript in Figure 3— figure supplement 1D.

Reviewer #2 (Recommendations for the authors):The authors should carefully check figure legends for reporting of individual mice and independent experiments for each panel. Frequently, 3 independent experiments with 3 mice each are reported and fewer dots for individual mice are shown. For instance, in figure 2, in some instances 4 dots per group are shown, which is more than an individual "representative" experiment but too few to be justified by elimination of outliers, as this would indicate more outliers than usable data points. The authors should check whether comparison of multiple groups requires ANOVA, such as in Figure 4A.

We apologize for the ambiguous labels and figure legends. We have rechecked all figure legends and reported samples for each panel individually. For Figure 4A, we apologize for the ambiguous label. Now we conducted pair-wise t-test between ETPs and other individual subsets, bars and stars were added for each of these tests.

The authors report higher levels of CD47 on ETPs, when compared to bone-marrow progenitors and later stages. However, ETPs are already in the thymus, whereas the progenitors should be expected to have higher levels. The authors should reassess their interpretation of data. They could also subdivide ETPs according to Flt3 expression into young and old ETPs. CD47 flow cytometry only shows a minor shift over isotype even for ETPs (4E figure supplement 1). The authors should elaborate on their normalization procedure for cytometry data and also show primary data examples for cell types analyzed in Figure 4A.

a) As suggested, we added Flt3 staining in FACS detection. Due to the continuous expression of Flt3 on ETPs, CD47 expression is compared between top ¼ (Flt3hi) and bottom ¼ (Flt3lo) subsets. We found higher CD47 expression on Flt3hi ETP subset, indicating that newly immigrated ETPs has higher level of surface CD47, suggesting the function of CD47 in thymic homing. We have added these data to new Figure 4B.

b) Normalization and raw data of CD47 flow cytometry on ETPs and TPECs have been added to new Figure 4—figure supplement 1G,H, where mean fluorescence intensity (MFI) of CD47 of each samples were shown alongside their histogram overlay in panel G, as well as net CD47 MFI calculated as “(antiCD47 signal) – (corresponding isotype signal)” in panel H.

c) In Figure 4A, raw data of CD47 flow cytometry of representative cell subsets (LSK, ETP, DP) and their corresponding isotype controls were shown in new Figure 4—figure supplement 1A.

For the experiments og thymus regeneration, it would strengthen the paper, if the authors performed additional homing assays such as those performed by Zlotoff et al., (2011) or used the conditional Tie2Cre-SIRPa-KO model.

Thank you for your suggestion. As has been detailed in our response to this Reviewer’s public review, we have done thymic regeneration assay directly in *Sirp_α_*^-/-^ and control mice using SL-TBI model, and found impaired DN thymocyte development, while later thymocyte development is normal (Author response image 3). In addition, we have also tested thymic progenitor homing and T cell development in non-irradiated *Sirp_α_*^-/-^ or control mice, and found impaired thymic homing in *Sirp_α_*^-/-^ mice and accompanied thymocyte development at early stage (Figure 2I-K).

**Author response image 3. sa2fig3:** Reconstitution of bone marrow-derived progenitors in *Sirp_α_^-/-^* mice. (A) Schematic view of the experiment. (B,C) Statistics of proportion (B) and cell number (C) of donor derived cells in the thymus 4 weeks after SL-TBI and adoptive transfer. n=6 in each group, unpaired t-test applied. *: p <0.05, **: p<0.01.

The manuscript should be carefully checked for grammatical errors.

We apologize for that and have carefully checked and corrected these errors.

Reviewer #3 (Recommendations for the authors):Overall this was a very well-conceived study with some important and novel findings. Although I do believe that the experiments detailed support the main conclusions of the manuscript, the following are a few questions/comments that would strengthen the manuscript if addressed.– The authors have elegantly shown the importance of TPECs in guiding the entry of HPCs into the thymus, and in their previous work the authors have shown that TPECs associate with perivascular spaces. However, given that not all TPECs express Sirpa (Figure 1E), and in their previous work there are at least 20% of TPECs that do not associate with PVS (and it is not at all clear from their previous work what the proportion of TPECs are located at the CMJ where we expect HPCs to enter the thymus), it would be extremely beneficial to show the location of TPECs and Sirpa in the thymus in relation to other architectural features (such as PVS, CMJ etc.)

Thank you very much for your advice. For comparison of SIRP_α_ expression in TPECs locating at various thymic structure, we conducted confocal microscopic detection of SIRP_α_ expression on TPECs (CD31^+^Ly6C^-^) with perivascular spaces (PVS) surrounding the EC cavity or not. Regions of TPECs were automatically identified by CD31-expression area in ImageJ, SIRP_α_ expression were then measured in the TPEC region. The expression of SIRP_α_ in TPECs with or without PVS structure was in similar level, suggesting SIRP_α_ expression may not be regulated by thymic microenvironment signal. This result has been presented as Figure 1—figure supplement 1A.

– In Figure 2 I appreciate the inclusion of BM progenitor subsets but it would be great if additional subsets could be retroactively included such as LMPPs. This is not critical but would strengthen the claims if possible.

We updated gates for CLP in Figure 2—figure supplement 1F as cKit^lo^ Sca-1^lo^ IL7Ra^+^ (gating strategy is shown in Author response image 4), and also added analysis for lymphoid myeloid progenitors (LMPP) as cKit^+^Sca-1^+^Flt3^hi^. We found no difference in cell number or proportion of LMPP of total bone marrow cells in *Sirp_α_*^-/-^ and control mice (Author response image 4,C).

**Author response image 4. sa2fig4:** Analysis of lymphoid progenitors in *Sirp_α_* mice. Gating strategy for common lymphoid progenitors (CLPs) and lymphoid myeloid progenitors (LMPPs). (B,C) Statistics of cell number (B) and proportion (C) of LMPPs in the bone marrow, n=3 in *Sirp_α_*^+/-^ and n=4 in *Sirp_α_*^-/-^. Unpaired t-test applied. n.s. not significant.

– In Figure 7, is there any change in number of TPECs in mice treated with CV1?

After SL-TBI (5.5Gy) or at steady state, mice were administrated with CV1 (200_µ_g) intraperitoneally 3 times every 4 days, TPECs are detected by FACS. The results showed that CV1 treatment did not change TPEC proportion or cell number at either state (Author response image 5)

**Author response image 5. sa2fig5:** Thymic EC composition upon SIRPα blockade. (A) Cell number of thymic endothelial cells in the thymus after indicated treatment. (B,C) Cell number (B) and proportion (C) of total thymic portal endothelia cells (TPECs) of a thymus. n=3 in each group. Unpaired t-test applied. n.s. not significant.

– As a stylistic note, specific figures/panels should not be called out in the discussion.

Thank you for your advice. We have modified the Discussion section.

References:

Jaiswal, S., Jamieson, C.H., Pang, W.W., Park, C.Y., Chao, M.P., Majeti, R., Traver, D., van Rooijen, N., and Weissman, I.L. (2009). CD47 is upregulated on circulating hematopoietic stem cells and leukemia cells to avoid phagocytosis. Cell *138*, 271-285.

Krueger, A. (2018). Thymus Colonization: Who, How, How Many? Archivum immunologiae et therapiae experimentalis *66*, 81-88.

Krueger, A., Willenzon, S., Łyszkiewicz, M., Kremmer, E., and Förster, R. (2010). CC chemokine receptor 7 and 9 double-deficient hematopoietic progenitors are severely impaired in seeding the adult thymus. Blood *115*, 1906-1912.

Prockop, S.E., and Petrie, H.T. (2004). Regulation of thymus size by competition for stromal niches among early T cell progenitors. Journal of immunology (Baltimore, Md. : 1950) *173*, 1604-1611.

Zlotoff, D.A., Sambandam, A., Logan, T.D., Bell, J.J., Schwarz, B.A., and Bhandoola, A. (2010). CCR7 and CCR9 together recruit hematopoietic progenitors to the adult thymus. Blood *115*, 1897-1905.

Zlotoff, D.A., Zhang, S.L., De Obaldia, M.E., Hess, P.R., Todd, S.P., Logan, T.D., and Bhandoola, A. (2011). Delivery of progenitors to the thymus limits T-lineage reconstitution after bone marrow transplantation. Blood *118*, 1962-1970.